# Dynamic large-scale connectivity of intrinsic cortical oscillations supports adaptive listening in challenging conditions

**Mohsen Alavash** [1,2]*, **Sarah Tune** [1,2], **Jonas Obleser** [1,2]*

1 Department of Psychology, University of Lübeck, Lübeck, Germany, 2 Center of Brain, Behavior, and Metabolism, University of Lübeck, Lübeck, Germany

* mohsen.alavash@uni-luebeck.de (MA); jonas.obleser@uni-luebeck.de (JO)

**Data Availability Statement:** The complete dataset associated with this work is publicly available under https://osf.io/ge2cq/.

## Abstract

In multi-talker situations, individuals adapt behaviorally to this listening challenge mostly with ease, but how do brain neural networks shape this adaptation? We here establish a long-sought link between large-scale neural communications in electrophysiology and behavioral success in the control of attention in difficult listening situations. In an age-varying sample of $N = 154$ individuals, we find that connectivity between intrinsic neural oscillations extracted from source-reconstructed electroencephalography is regulated according to the listener's goal during a challenging dual-talker task. These dynamics occur as spatially organized modulations in power-envelope correlations of alpha and low-beta neural oscillations during approximately 2-s intervals most critical for listening behavior relative to resting-state baseline. First, left frontoparietal low-beta connectivity (16 to 24 Hz) increased during anticipation and processing of a spatial-attention cue before speech presentation. Second, posterior alpha connectivity (7 to 11 Hz) decreased during comprehension of competing speech, particularly around target-word presentation. Connectivity dynamics of these networks were predictive of individual differences in the speed and accuracy of target-word identification, respectively, but proved unconfounded by changes in neural oscillatory activity strength. Successful adaptation to a listening challenge thus latches onto two distinct yet complementary neural systems: a beta-tuned frontoparietal network enabling the flexible adaptation to attentive listening state and an alpha-tuned posterior network supporting attention to speech.

## Introduction

Noisy, multi-talker listening situations make everyday communication challenging: how to focus only on what we want to hear? Behavioral adaptation to a listening challenge is often facilitated by listening cues (e.g., spatial location or semantic context) and requires individual cognitive ability to control attention [1,2]. How do a listener's brain networks shape this behavioral adaptation? Addressing this question is essential to neurorehabilitation of the hearing impaired or to the advancement of aided hearing [3,4].

**Funding:** Research was supported by the European Research Council (ERC Consolidator grant AUDADAPT, no. 646696 to JO; https://cordis. europa.eu/project/id/646696) and German Research Foundation (DFG grant, no. AL2408/1-1 to MA; https://gepris.dfg.de/gepris/projekt/ 426620060?context=projekt&task=showDetail&id= 426620060&). The funders had no role in study design, data collection and analysis, decision to publish, or preparation of the manuscript.

**Competing interests:** The authors have declared that no competing interests exist.

**Abbreviations:** AMI, attentional modulation index; BF, Bayes factor; BIC, Bayesian information criterion; FDR, false discovery rate; FIR, finite impulse response; fMRI, functional magnetic resonance imaging; FS, full scale; HCP, Human Connectome Project; ICA, independent component analysis; M/EEG, magneto/ electroencephalography; MRI, magnetic resonance imaging; OR, odds ratio; PCC, partial canonical coherence; PTA, pure-tone audiometry; 6CIT, 6-Item Cognitive Impairment Test.

Our recent study provided a large-scale brain network account of successful listening [5]. Using functional magnetic resonance imaging (fMRI), we measured participants' brain hemodynamic activity during task-free resting state and a challenging speech comprehension task. We were able to explain individual adaptation to the listening challenge by reconfiguration of an auditory-control brain network toward increased modular segregation during attentive listening. Knowing the indirect relationship between brain hemodynamics and neural oscillatory dynamics [6–8], our study posed an important underexplored question: whether and how network interactions between intrinsic neural oscillations change to support behavioral adaptation to a listening challenge?

Magneto-/electroencephalography (M/EEG) studies on attentive listening provide ample evidence supporting the role of neural oscillatory activity within the alpha band (approximately 8 to 12 Hz) in top-down attentional control over incoming auditory streams [9]. Specifically, attentional modulation of alpha-band power has been reported extensively when listeners selectively attend to 1 of 2 (or more) concurrent sounds [10–16]. These findings have been often interpreted in the light of the widely recognized inhibitory role of alpha oscillations as top-down modulation of cortical excitability [17,18].

However, these studies have usually investigated how selective attention modulates the power of neural activity and assume cortical alpha rhythms in temporal or parietal regions to underlie these dynamics [19]. Neurophysiological studies on visuospatial attention provide substantial guiding evidence on how the functionally specialized coupling between distant cortical nodes involving prefrontal regions underlies the dynamics of attentional selection and control ([20–22]; see [23] for review). Nevertheless, we do not know yet whether and how attentive listening relies on coupling between neural oscillations across distributed cortical nodes. We here address this question within the unifying framework of intrinsic coupling modes by focusing on large-scale neural interactions that arise from coherent aperiodic fluctuations in signal envelopes that unfold at slow time scales (<0.1 Hz) [24].

Specifically, in M/EEG recordings, envelope fluctuations of ongoing alpha- and beta-band oscillations (approximately 8 to 32 Hz) exhibit these dynamics, and their intrinsic cofluctuations over time correlate with covariations in brain hemodynamic activity during resting state [7,8,25,26]. Accordingly, these neural dynamics have been proposed as one network mechanism whereby distant cortical regions functionally coordinate their activity to support performance in sensorimotor or cognitive tasks [24,27,28]. In line with this proposal, coherent neural activity during task states has been shown to subserve multisensory perception, attention, and cognitive control [29,30]. How intrinsic neural oscillations regulate their envelope coupling to support behavioral adaptation to attentive listening is unknown.

Moreover, it is not yet fully understood why listeners exhibit substantial interindividual variability in successful adaptation to a listening challenge [31,32]. This variability seems to arise from differences in sensory coding fidelity, from differences in the ability to use cognitive resources, or from a combination of both [16,33]. To what degree this interindividual variability relates to listeners' ability to top-down regulate amplitude-coupling between neural oscillations is unknown.

The present large-sample EEG study regards the neocortical systems involved in attentive listening as an assembly of dynamic large-scale networks of ongoing neural oscillations. Building on our previous work on brain hemodynamic networks during the same experimental paradigm [5], we treat the resting-state network makeup of intrinsic neural oscillations as their putative task network at its "idling" baseline. We predict neural oscillations to regulate their amplitude-coupling and/or reconfigure their network in adaptation to attentive listening. We

expect these changes to manifest in the alpha–beta frequency range and leverage the degree of these changes as a proxy of individuals' successful adaptation to a listening challenge.

## Results

We recorded 64-channel scalp EEG from a large, age-varying sample of healthy middle-aged and older adults ($N = 154$; age range = 39 to 80 y, median age = 61 y; 62 males). This includes 30 individuals who have had participated in our previous fMRI study and performed the same listening task [5]. All participants were recruited as part of an ongoing large-scale longitudinal study (see Data collection for details). Each participant completed a 5-min eyes-open resting state measurement and 1 h of the listening task. The listening task was identical to the one used in [5] and can be viewed as a linguistic Posner paradigm (Fig 1).

In brief, participants were dichotically presented with two 5-word sentences spoken by the same female talker and were instructed to identify the final word (i.e., target) of one of these 2 sentences. To probe individual use of auditory spatial attention and semantic prediction when confronted with a listening challenge, sentence presentation was preceded by 2 visual cues. First, a spatial-attention cue either indicated the to-be-attended side, thus invoking selective attention, or it was uninformative, thus invoking divided attention. The second cue informed

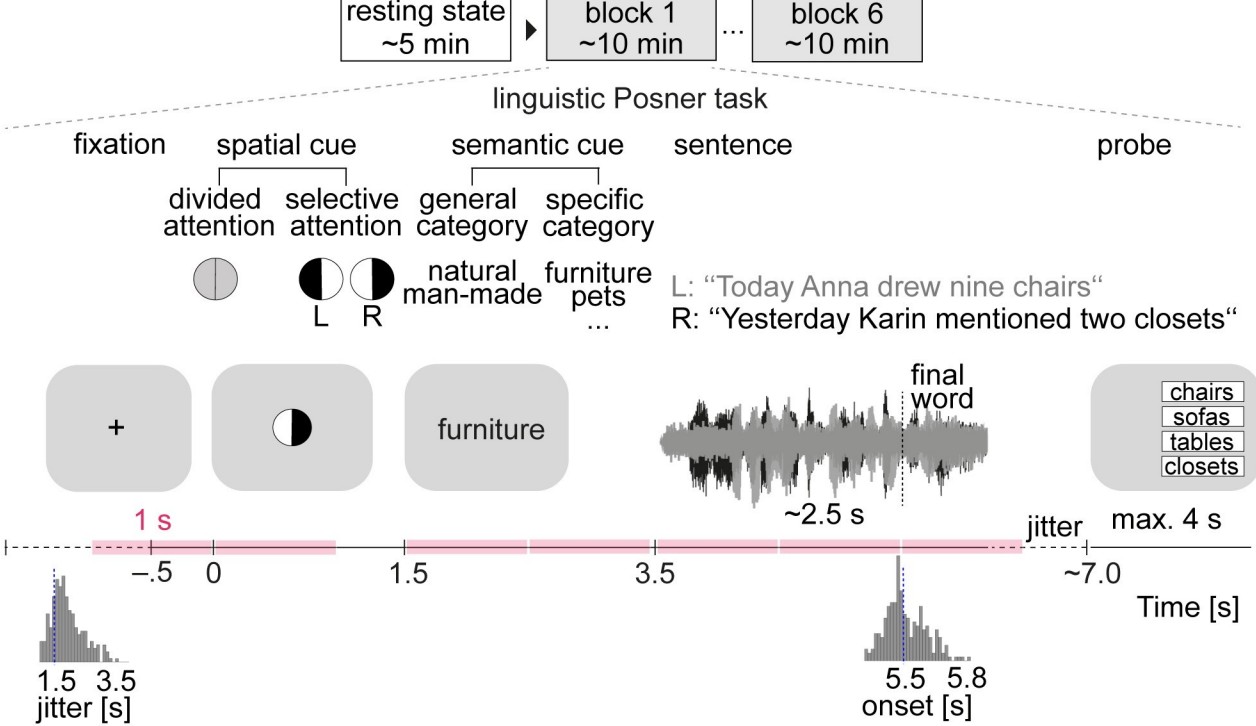

**Fig 1. Experimental procedure and the listening task.** EEG of $N = 154$ participants was recorded during a 5-min eyes-open resting state and 6 blocks of a linguistic Posner task with concurrent speech [5]. Participants listened to 2 competing, dichotically presented sentences. Each trial started with the visual presentation of a spatial cue. An informative cue provided information about the side (left ear vs. right ear) of the to-be-probed final word. An uninformative cue did not provide information about the side of the to-be-probed final word. A semantic cue was visually presented indicating a general or a specific semantic category for both final words. The 2 sentences were presented dichotically along with a visual fixation cross. At the end of each trial, a visual response array appeared on the side of the probed ear with 4 word choices, asking participants to identify the final word of the sentence presented to the respective ear. To capture amplitude-coupling between frequency-specific neural oscillations throughout the listening task, power-envelope correlations between narrow-band EEG source signals were estimated for 1-s time windows of interest (colored intervals) and compared with resting state connectivity at the same frequency. The stimulus materials can be found at https://osf.io/nfv9e/. EEG, electroencephalography.

about the semantic category of both final words either very generally or more specifically, allowing for more-or-less precise semantic prediction of the upcoming target word.

We source-localized narrow-band EEG signals recorded during rest and task, and following leakage correction, estimated power-envelope correlations between all pairs of cortical regions defined according to a symmetric whole-brain parcellation template (S1 Fig). We asked whether and how amplitude-coupling between intrinsic neural oscillations changes throughout the listening task as compared to resting state. Importantly, using (generalized) linear mixed-effects models, we examined the influence of listening cues and frequency-specific network dynamics on single-trial listening performance, accounting for individuals' age, hearing thresholds, and neural oscillatory power.

## Informative cues improve listening success

The analysis of listening performance using linear mixed-effects models revealed an overall behavioral benefit from more informative cues. The behavioral effects reported below are in good agreement with the results obtained before using the same task in fMRI [5]. Specifically, listeners performed more accurately and faster under selective attention as compared to divided attention (accuracy: odds ratio (OR) = 3.4, $p < 0.001$; response speed: β = 0.57, $p < 0.001$; Fig 2A and 2B, top row of scatter plots). Moreover, listeners performed faster when they were cued to the specific semantic category of the final word as compared to a general category (β = 0.2, $p < 0.001$; Fig 2B, second scatter plot). We did not find evidence for any interactive effects of the 2 listening cues in predicting accuracy (OR = 1.3, $p = 0.15$) or response speed (β = 0.09, $p = 0.29$).

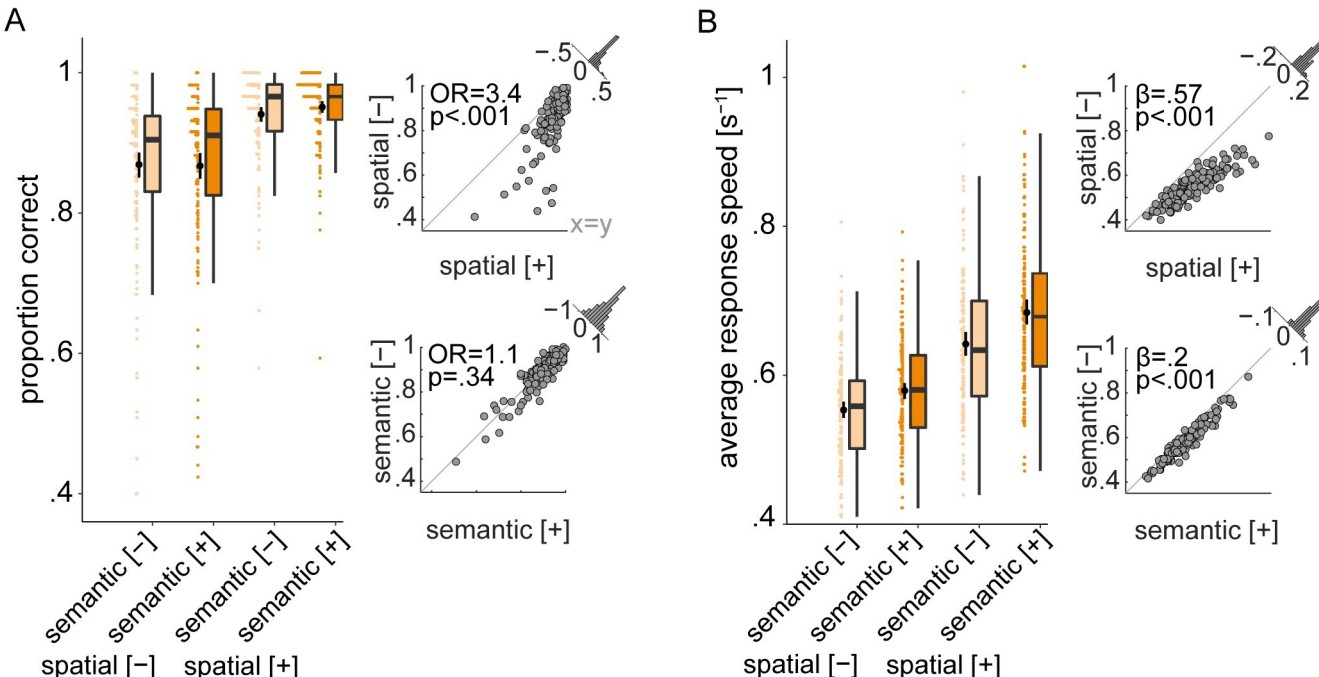

**Fig 2. Individual behavioral benefit from informative listening cues. (A)** Proportion of correct final word identifications averaged over trials per cue condition **(B)** The same as (A) but for average response speed. Box plots: Colored data points represent trial-averaged performance scores of $N = 154$ individuals per cue–cue combination. Black bars show mean ± bootstrapped 95% CI. Scatter plots: Individual cue benefits shown separately for each cue and performance score. Black data points represent individuals' trial-averaged scores under informative [+] and uninformative [−] cue conditions. Gray diagonal corresponds to 45-degree line. Histograms show the distribution of the cue benefit (informative minus uninformative) across all participants. OR: odds ratio parameter estimate resulting from generalized linear mixed-effects models; β: slope parameter estimate resulting from general linear mixed-effects models. The data underlying this figure can be found at https://osf.io/ge2cq/.

As expected, the older a listener the worse the performance (main effect of age; accuracy: OR = 0.78, $p < 0.01$; response speed: β = −0.15, $p < 0.001$). Furthermore, as to be predicted from the right-ear advantage for linguistic materials [34,35], listeners were more accurate and faster when probed on the right compared to the left ear (main effect of probe on accuracy: OR = 1.25, $p < 0.01$; response speed: β = 0.09, $p < 0.001$). In addition, the number of blocks participants completed (6 in total) had a main effect on task performance (accuracy: OR = 1.28, $p < 0.001$; response speed: β = 0.11, $p < 0.001$) indicating that individuals' listening performance improved over task blocks (see S1 and S2 Tables for all model terms and estimates).

## Spectral, spatial, and topological profile of power-envelope correlations under rest and listening

Central to the present study, we asked whether and how amplitude-coupling between intrinsic neural oscillations changes as individuals engage in attentive listening. To answer this question, we first investigated the spectral and spatial profile of power-envelope correlations estimated under rest and the listening task.

In line with previous studies (e.g., [8,25,36]), power-envelope correlations were strongest within 7 to 24 Hz (Fig 3A) and proved reliable in this range under both rest and task conditions (between-subject analysis; see S2 Fig for details). In addition, across 30 individuals who had participated in both the present EEG study and our previous fMRI study [5], EEG mean connectivity showed consistently positive correlations with fMRI mean connectivity within the same frequency range (Fig 3B).

This frequency range overlaps with the broad-band alpha (7 to 14 Hz) and the lower end of beta band (16 to 24 Hz). The dominant frequency of alpha-band oscillatory activity appears to

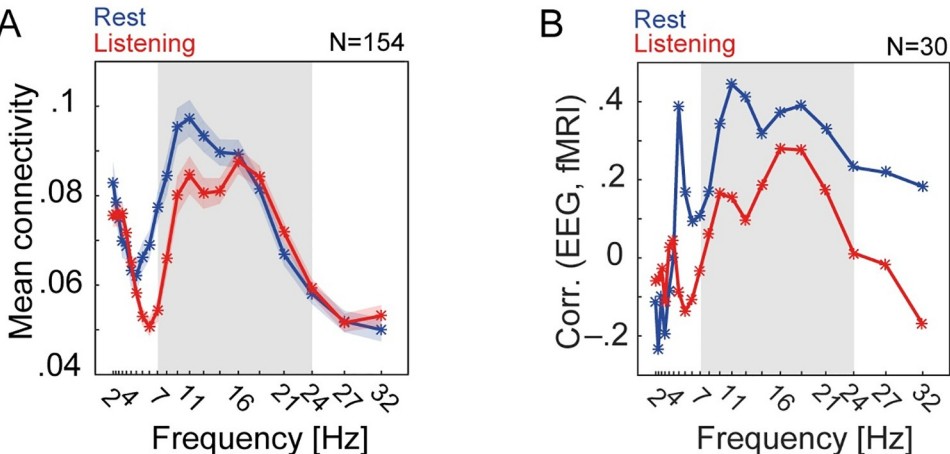

**Fig 3. Spectral profile of source EEG power-envelope correlations and their relationship with fMRI connectivity.**
**(A)** Frequency-specific whole-brain mean connectivity during rest and listening task. At each frequency and per individual, mean connectivity was estimated as the upper-diagonal average of power-envelope correlation matrix thresholded at 10% of network density. Each line graph represents mean ± SEM of mean connectivity across all $N =$ 154 participants. **(B)** Thirty participants have had performed the same task in fMRI [5]. Across these participants, we tested the correlation between whole-brain mean connectivity derived from brain hemodynamics (i.e., mean Pearson correlation between fMRI band-passed filtered [0.06–0.12 Hz] BOLD signals) and whole-brain mean connectivity derived from EEG oscillatory source activity (i.e., mean power-envelope correlation at frequencies ranging from 2–32 Hz). Power-envelope correlations between EEG oscillatory sources show strongest positive correlation with fMRI connectivity within alpha and low-beta frequency range (gray frequency intervals). The data underlying this figure can be found at https://osf.io/ge2cq/. EEG, electroencephalography; fMRI, functional magnetic resonance imaging.

vary across cortex and task conditions [37,38]. Specifically, while alpha oscillations at the lower end of the frequency range (7 to 11 Hz) have been recorded from occipitoparietal sources during rest or visual tasks and temporal sources during auditory tasks, these oscillations appear slightly faster over sensorimotor cortices [37,39–41]. Spectral modes as such have been shown to be characteristic to individual cortical areas and organize—according to their similarity—into large-scale networks [37]. Accordingly, we focused our main analysis on three frequency bands within 7 to 24 Hz, namely $\alpha_1$ (7 to 11 Hz), $\alpha_2$ (11 to 14 Hz), and $\beta_1$ (16 to 24 Hz). The main analysis began by investigating group-average whole-brain connectivity per frequency band under rest and listening task.

As shown in Fig 4, rest and task connectivity showed overall similar spatial profiles across cortex. More precisely, power-envelope correlations in both $\alpha_1$ and $\alpha_2$ range were strongest within and between bilateral occipital and temporal nodes, but also across occipitoparietal as well as temporoparietal nodes under rest or listening task (Fig 4, first column; see S4 Fig for $\alpha_2$ connectivity). In contrast, frontal nodes displayed relatively sparse connectivity. Power-envelope correlations in $\beta_1$ frequency range displayed similar spatial pattern, but with the left-hemispheric connections showing relatively stronger connectivity under listening task as compared to rest (Fig 4, second column).

To evaluate connectivity strength of each cortical node, we used a simple graph-theoretical measure—namely, nodal connectivity—defined as the sum of each node's connection weights. This analysis revealed a long-tailed distribution of nodal connectivity across cortex, such as nodes with highest connectivity strength localized to occipital and temporal regions in both the alpha and low-beta frequency range (Fig 4, cortical maps).

We next investigated the modular topology of these correlation structures. Modularity of a large-scale network refers to its decomposability into smaller subnetworks. The notion stems from contemporary graph theory, and it has been extensively applied to fMRI connectivity data in recent years. Specifically, it has been shown that brain regions functionally organize into clusters that are relatively densely *intra*-connected but sparsely *inter*-connected [42]. This topological organization is thought to represent functional segregation across the human cortex [43]. Modularity of a network can be estimated by the Newman optimization algorithm using the so-called modularity index [44,45]: The more modular the network, the closer the index to 1. To what degree the correlation structures of alpha or low-band neural oscillations show similar organization is not known. In our previous fMRI study, brain networks had clearly displayed this organization under both rest and the same listening task [5]. Accordingly, we predicted EEG source connectivity to be topologically decomposable into network modules.

However, we found that neither alpha nor low-beta connectivity displayed modular organization under rest or listening: The module detection algorithm revealed only unevenly dense modules and relatively small modularity indices (S3 Fig). Collectively, these results illustrate that power-envelope correlations in the alpha and low-beta frequency range were predominantly present across sensory and parietal association regions. However, envelope-coupling between these regions did not topologically segregate to exhibit a modular organization.

## Hyperconnectivity of frontoparietal β oscillations in anticipation of and during spatial cueing

To investigate whether and how power-envelope correlations between neural oscillations change throughout the listening task, we leveraged the high temporal resolution of EEG and defined 1-s time windows of interest throughout the entire trial (Fig 1, colored intervals). This allowed us to estimate connectivity for each frequency band and time window by

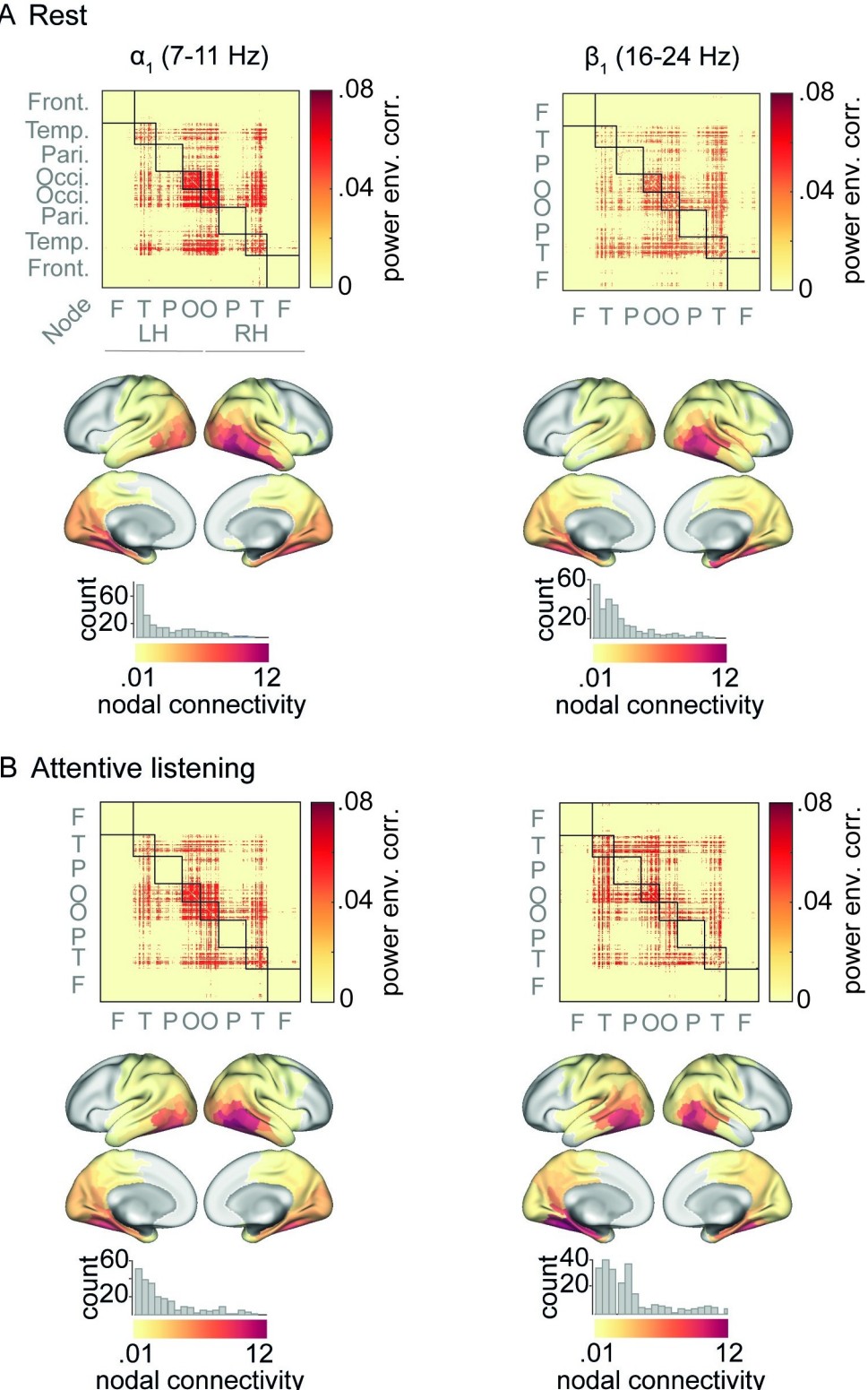

**Fig 4. Connectivity maps of α/β oscillations under rest and attentive listening. (A)** For each frequency band, power-envelope correlations between EEG oscillatory sources were estimated using 5-min eyes-open resting state data. **(B)** The same procedure as in (A) was applied to task data after concatenating whole-trial signals across 30 random trials of each block (5-min data in total as in rest), and then averaging the correlation matrices across all 6 blocks of

task. Correlation matrices were averaged across $N = 154$ individuals and thresholded at 10% of network density. Nodes having zero connectivity strength are masked in gray. Histograms illustrate distribution of nodal connectivity strength with high-connectivity nodes overlapping with occipital and posterior temporal regions. Note that in this analysis, task connectivity is not specific to a particular time window within trials or cue condition, and thus illustrate the overall spatial profile of $\alpha/\beta$ connectivity under listening task. Nodes correspond to cortical parcels as in [46] and are grouped according to their cortical lobes. The data underlying this figure can be found at https://osf.io/ge2cq/. EEG, electroencephalography; LH, left hemisphere; RH, right hemisphere.

concatenating windowed narrow-band signals across all 240 trials (equivalently, 4-min data). We then compared the results with 4-min resting state connectivity at the same frequency band by subtracting task and rest connectivity matrices.

During anticipation of the spatial-attention cue (−1 to 0 s), $\beta_1$ connectivity showed an increase relative to resting state (Fig 5A, left panel, first connectivity matrix). This hyperconnectivity was observed most prominently within the left temporal cortex, as well as between frontal, temporal, and parietal regions. Using permutation tests to compare nodal connectivity between rest and task, we found that $\beta_1$ connectivity of mainly left prefrontal and parietal nodes was increased during anticipation of the spatial cue relative to rest (0.29 < Cohen's d < 0.46, $p < 0.01$, false discovery rate (FDR)-corrected for multiple comparisons across nodes; significant nodes are outlined in black in Fig 5A, first panel, first cortical maps). In contrast, alpha connectivity during this period was not significantly different from resting state (S5 Fig).

Next, we focused on 1-s time windows capturing the presentation of each listening cue. During the spatial cue presentation (0 to 1 s), $\beta_1$ connectivity of left prefrontal and parietal regions remained significantly higher than rest (0.29 < d < 0.44, $p < 0.01$; Fig 5A, first panel, second connectivity matrix; see S4 Fig for nondifferential connectivity matrices). During the same period, alpha-band connectivity showed a relative decrease, which was only significant in $\alpha_1$ range and over a few parietal nodes (S5 Fig). During the semantic cue period (1.5 to 2.5 s) power-envelope correlations were not significantly different from resting state in neither the alpha nor in low-beta range (S5 Fig).

To assess the degree and direction to which each listener showed $\beta_1$ hyperconnectivity during presentation of the spatial cue, we averaged nodal connectivity across frontoparietal regions per individual. We then statistically compared individuals' mean connectivity values between rest and task using a permutation test. In accordance with the results at the nodal level, mean frontoparietal $\beta_1$ connectivity was significantly higher than rest during presentation of the spatial cue (d = 0.6, $p < 0.01$; Fig 5B, first panel).

## Hypoconnectivity of posterior cortical α oscillations during listening to speech

During the listening task, participants were asked to identify the final word of one of two concurrent sentences. The sentences were presented after the two listening cues and, on average, had a duration of around 2.5 s (Fig 1). Accordingly, we divided the sentence presentation period (3.5 to 6.5 s) into three consecutive 1-s intervals and estimated connectivity for each frequency band and time window. This analysis revealed a gradual decrease in alpha connectivity across posterior parts of the brain during sentence presentation relative to resting state.

Specifically, $\alpha_1$ connectivity within and between bilateral occipital regions was decreased relative to resting state, particularly so during the interval around the final-word presentation (Fig 5A, second panel). This hypoconnectivity was also observable between occipital and temporoparietal regions and in the $\alpha_2$ band (S6 Fig). In contrast, $\beta_1$ connectivity during sentence presentation was not significantly different from resting state (S6 Fig, third column).

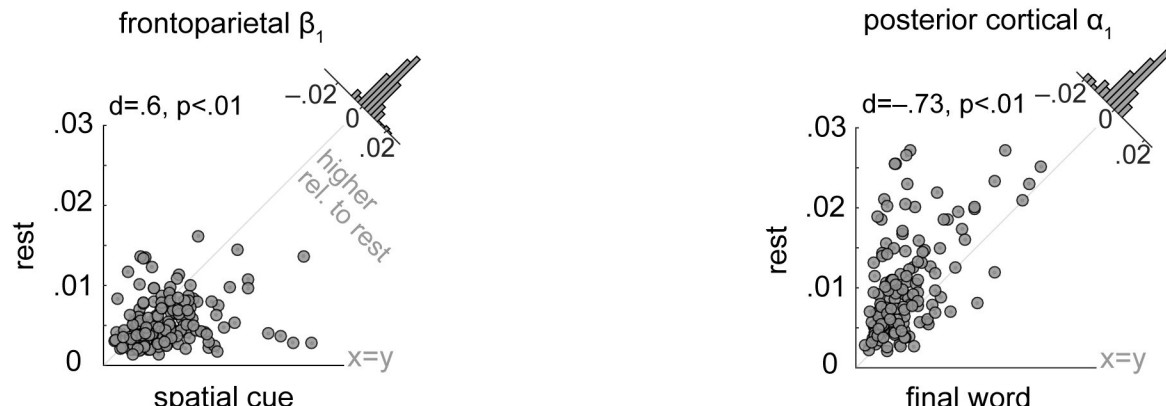

## A Nodal connectivity

## B Mean connectivity

**Fig 5. Cortical connectivity dynamics of α/β oscillations during the listening task. (A)** To assess whether and how intrinsic alpha and low-beta oscillations regulate their cortical connectivity during attentive listening, connectivity difference maps (i.e., task minus rest) were derived per frequency band. Task connectivity was estimated by concatenating 1-s windowed signals across all 240 trials (4-min data). In anticipation of and during the spatial cue presentation, $\beta_1$ connectivity was increased relative to its intrinsic connectivity mainly within the left hemisphere (first panel). This hyperconnectivity was significant across frontoparietal regions (brain surfaces; significant nodes are outlined in black). The same analysis revealed a significant $\alpha_1$ hypoconnectivity during final-word presentation (second panel). Nodes correspond to cortical

parcels as in [46] and are grouped according to their cortical lobes. **(B)** To assess the degree and direction of change in connectivity per individual listener, nodal connectivity was averaged across significant nodes and compared between rest and task using permutation tests. Data points represent individuals' mean connectivity ($N$ = 154). Gray diagonal corresponds to 45-degree line. Histograms show the distribution of the connectivity change (task minus rest) across all participants. The data underlying this figure can be found at https://osf.io/ge2cq/. dlPFC, dorsolateral prefrontal cortex; FEF, frontal eye field; IFG, inferior frontal gyrus; IPL, inferior parietal lobule; LH, left hemisphere; PSL, perisylvian language area; RH, right hemisphere; STS, superior temporal sulcus.

When tested at the nodal level, the alpha-band hypoconnectivity overlapped with posterior temporal cortices as well as occipital and parietal regions ($-0.5 < d < -0.23$, $p < 0.01$; nodes outlined in black in Fig 5A, second panel, cortical maps). Notably, when averaged across the posterior cortical nodes, individuals showed significantly lower mean connectivity during final-word presentation relative to rest (Fig 5B, second panel).

## Connectivity dynamics of intrinsic α/β oscillations predict individual listening behavior

The results illustrated above can be outlined as spatiotemporal modulations in alpha and low-beta connectivity throughout the listening task relative to resting state. As Fig 5B shows, listeners clearly showed interindividual variability in the degree and direction of these connectivity dynamics. Thus, we next investigated whether these variabilities could account for interindividual variability in listening performance (Fig 2).

To this end, we tested the direct and interactive effects of individual mean connectivity during resting-state and during each listening task interval, i.e., spatial-cue or final-word periods, on individuals' accuracy or response speed. These brain-behavior relationships were tested in separate models per low-beta and alpha band. In each model, brain regressors were the mean connectivity of the frontoparietal low-beta or posterior alpha networks on an individual level, respectively. These are the same linear mixed-effects models based on which the beneficial effects of informative listening cues on behavioral performance was reported earlier (Fig 2).

Fig 6 illustrates the main brain-behavior results. First, we found that frontoparietal $\beta_1$ connectivity during spatial cueing predicted listeners' response speed in the ensuing final-word identification, but only in those individuals whose resting-state $\beta_1$ connectivity of the same network was lower than average (by half-SD or more). Statistically, this surfaced as a significant interaction between mean resting-state frontoparietal $\beta_1$ connectivity and mean connectivity of the same network during spatial-cue presentation in predicting listeners' response speed ($\beta = 0.019$, $p < 0.01$; Fig 6A; see S6 Table for details). This indicated that, among individuals with lower-than-average resting-state frontoparietal $\beta_1$ connectivity ($N = 41$), increased connectivity during spatial cueing was associated with slower responses during final-word identification ($\beta = -0.034$, $p < 0.01$; Fig 6A, first panel). The main effect of frontoparietal $\beta_1$ connectivity on response speed was not significant (resting state: $\beta = -0.017$, $p = 0.74$; spatial cue: $\beta = -0.01$, $p = 0.18$).

Second, mean posterior $\alpha_1$ resting-state connectivity and mean connectivity of the same network during final-word presentation jointly predicted a listener's accuracy (OR = .94, $p = 0.04$; Fig 6B; see S1 Table for details). This interaction indicated that the behavioral relevance of $\alpha_1$ hypoconnectivity during final-word identification was conditional on individuals' resting-state $\alpha_1$ connectivity: for listeners with an intrinsic $\alpha_1$ connectivity higher than average (by half-SD or more; $N = 45$), there was a significant negative correlation between $\alpha_1$ connectivity during final-word presentation and overall word identification accuracy (OR = 0.88, $p = 0.04$; Fig 6B, third panel). The main effect of posterior $\alpha_1$ connectivity on accuracy was not significant (resting state: OR = 0.91, $p = 0.2$; spatial cue: OR = 0.98, $p = 0.69$).

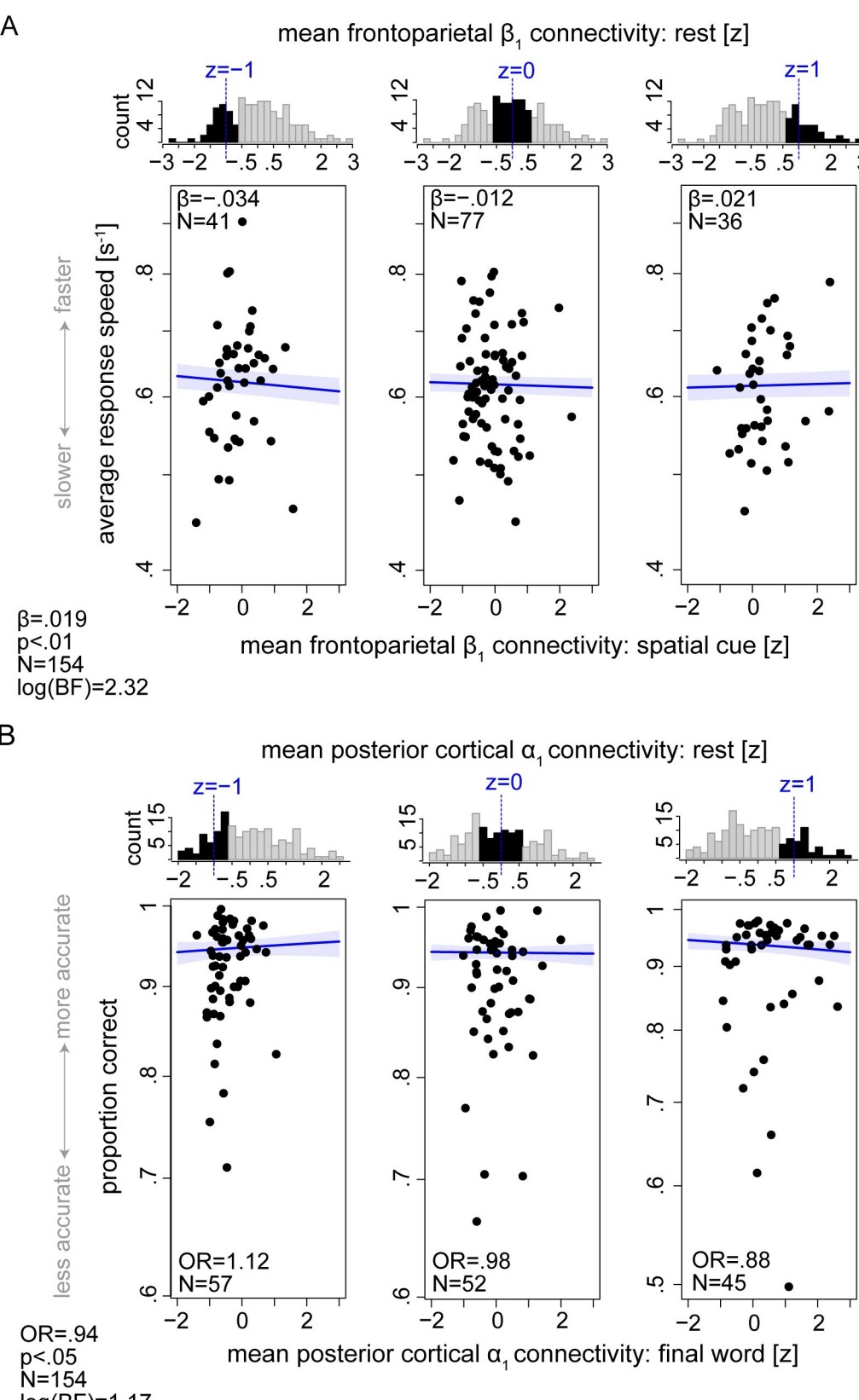

**Fig 6. Prediction of individual listening behavior from α/β connectivity dynamics. (A)** Interaction between mean frontoparietal $\beta_1$ connectivity derived from resting state and connectivity of the same network during spatial cue presentation predicted how fast listeners identified the final word in the ensuing sentence presentation ($\beta = 0.019$, $p < 0.01$, log-BF = 2.32). For visualization purpose only, individuals were grouped according to their standardized mean frontoparietal $\beta_1$ resting state connectivity. Each scatter plot corresponds to one group of individuals. Distribution of mean connectivity for each group is highlighted in black within the top histograms. Black data points represent the same individuals' trial-average response speed regressed on their standardized mean frontoparietal $\beta_1$ connectivity during spatial cueing. Solid blue lines indicate linear regression fit to the data when $\beta_1$ resting state connectivity held constant at the group mean (dashed blue line in histograms). **(B)** Interaction between mean posterior $\alpha_1$ connectivity derived from resting state and connectivity of the same network during final word presentation predicted listeners' word identification accuracy (OR = 0.94, $p < 0.05$, log-BF = 1.17). Data visualization is the same as in (A), but, here, the grouping variable is mean posterior $\alpha_1$ resting state connectivity and the predictor is the mean connectivity of the same network during final word period. Individuals' age and hearing thresholds have been accounted for in the models. Shaded area shows two-sided parametric 95% CI. $\beta$: Slope parameter estimates from linear mixed-effects model. OR: odds ratio parameter estimates from generalized linear mixed-effects models. The data underlying this figure can be found at https://osf.io/ge2cq/. BF, Bayes factor; OR, odds ratio.

Additionally, in separate models, we tested the above interactions using connectivity during anticipation of the spatial cue (−1- to 0-s fixation) as baseline (i.e., the classical contrast in event-related paradigms) instead of resting state connectivity. Notably, the $\beta_1$ interaction in predicting listeners' response speed was absent ($\beta = 0.001$, $p = 0.89$), as was the $\alpha_1$ interaction in predicting listeners' accuracy (OR = 0.98, $p = 0.5$).

We also investigated the reproducibility and robustness of the brain-behavior findings. To this end, we randomly split the data into 5 nonoverlapping folds and examined the strength and significance of the model's parameter estimate when subsets of data were used. In addition, to assess the relative strength of evidence (rather than statistical significance) in support of the hypothesis that the observed data are better explained by the model including the respective interaction terms, we calculated the Bayes factor (BF). By convention, a log-BF of 1 begins to lend support to the H1 [47].

The results obtained for the model predicting response speed from frontoparietal $\beta_1$ connectivity dynamics were well in line with those obtained from the full sample (S10 Fig): For all 5 model iterations, the β estimate of the interaction term was significantly larger than 0, and the log-BF estimate was larger than 1 in 3 model iterations. For the model predicting accuracy from posterior $\alpha_1$ connectivity dynamics, for 4 model iterations, the OR was significantly smaller than 1, indicating a negative correlation, and the log-BF estimate was larger than 1 in one model iteration. Overall, these results document within-sample reproducibility and robustness of the effects and support the more complex model that includes the interaction term.

In support of the behavioral relevance of alpha and low-beta connectivity, we also found their significant interactions with task-block number in predicting individuals' accuracy ($\alpha_1$: OR = 0.91, $p < 0.01$; $\alpha_2$: OR = 0.92, $p < 0.01$; $\beta_1$: OR = 0.95, $p = 0.02$) or response speed ($\alpha_1$: β = −0.021, $p < 0.001$; $\alpha_2$: β = −0.032, $p < 0.001$; see S1–S5 Tables for details). These interactions indicated that (1) those listeners with lower posterior alpha connectivity during final-word presentation showed improved behavioral performance over 6 blocks of task; and (2) listeners with lower $\beta_1$ connectivity during presentation of the spatial cue showed improved accuracy over task blocks.

We did not find any evidence for the behavioral relevance of the $\alpha_2$ band connectivity dynamics, i.e., its interaction with resting state connectivity (see S3 and S4 Tables for details). The interactions between mean connectivity or neural oscillatory power with the listening cues in predicting behavior were also not significant in neither of the frequency bands.

As mentioned earlier, in our brain-behavior models, we controlled for participants' chronological age and hearing thresholds. In additional analyses, we explored the possible correlations

between either of these two covariates with mean resting state or task connectivity per frequency band. None of these correlations were significant (correlations with age; $\alpha_{1,rest}$: Spearman's ρ = 0.01, $p$ = 0.89; $\alpha_{1,task}$: ρ = −0.01, $p$ = 0.99; $\beta_{1,rest}$: ρ = 0.02, p = 0.8, $\beta_{1,task}$: ρ = -0.1, $p$ = 0.2).

## Control analysis: Are α/β connectivity effects confounded by changes in activation?

A recurring theme in studying neural interactions is to what extent connectivity estimates are mere reflection of neural activation [48,49]. As an extreme scenario, power-envelope cofluctuations between two regions could be purely driven by instantaneous changes in neural activity (or signal-to-noise ratio) of each region and not their genuine functional connectivity. We therefore conducted a set of control analyses to assess the degree to which the dynamics of power-envelope correlations found here could be mere reflection of changes in alpha or low-beta power.

As illustrated in S7 Fig, grand-average power of narrow-band EEG signals during anticipation of the spatial cue was higher than whole-trial baseline within 7 to 24 Hz, and it was decreased in response to both listening cues. This modulation was similarly observed during sentence presentation and appeared stronger during final-word presentation. For the final-word period, we also quantified hemispheric lateralization of alpha-band activity (7 to 14 Hz) based on selective-attention trials and using the so-called attentional modulation index (AMI); AMI = (α-power$_{attendL}$ − α-power$_{attendR}$) / (α-power$_{attendL}$ + α-power$_{attendR}$). As expected, source alpha power was lower over the hemisphere contralateral to the side of attention (S7 Fig). At first glance, these changes in power appear similar to the power-envelope correlation dynamics reported earlier. To directly investigate this, we conducted the following analyses.

First, for each cortical node and across participants, we tested the correlation between nodal connectivity and mean nodal power (dB) during task. During the spatial cue period and within $\beta_1$ band, this correlation was mainly positive across cortex and was significant over a few occipital nodes (ρ = 0.3, $p$ < 0.01; S8 Fig). However, this relationship and the overall pattern of nodal correlations did not resemble the mainly left-hemispheric frontoparietal $\beta_1$ hyperconnectivity (Fig 5A, first panel). Moreover, during final-word presentation, the correlation between mean alpha power (dB) and nodal connectivity showed both positive and negative trends across cortex, which did not reach significance level (−0.25 < ρ < 0.25, $p$ > 0.1; S8 Fig, cf. Fig 5A, second panel). When these correlations were tested using mean connectivity and power averaged across frontoparietal or posterior nodes, there was only a positive trend for the correlation between mean frontoparietal $\beta_1$ connectivity and power during spatial-cue presentation (ρ = 0.15, $p$ = 0.07; S8 Fig).

Second, on selective-attention trials and during final-word presentation, alpha activity clearly showed a lateralized modulation depending on whether participants attended to left or right ear (S7 Fig). We investigated whether connectivity showed a similar lateralization. Precluding the here observed neural connectivity from being confounded by neural activity level, power-envelope correlations were not influenced by attentional-cue conditions (i.e., left versus right or selective versus divided) in neither of the trial intervals or frequency bands (S9 Fig).

Third, in our mixed-effects models, we included mean neural oscillatory power as a separate regressor and specifically tested the direct and interactive effects of power on individuals' behavioral performance. We only found a significant main effect of $\beta_1$ power during spatial-cue presentation on listeners' response speed (β = −0.029, $p$ < 0.01). Notably, while posterior $\alpha_1$ connectivity predicted listeners' behavioral accuracy, the data held no evidence for any effect of alpha power on behavior (see S1–S4 Tables for details). Additionally, while alpha and low-beta mean connectivity showed significant interactions with task-block number in predicting individuals' behavioral performance, this effect was absent for power.

Lastly, to diminish common covariation in source power due to volume conduction, all time-frequency source estimates in our analyses were orthogonalized across all pairs of cortical nodes prior to estimating power-envelope correlations [25]. This approach eliminates (or at least diminishes considerably), the instantaneous zero- or close to zero-lag correlations between signals [50]. Thus, connectivity estimates derived from this procedure are less likely to be contaminated by shared neural activity time-locked to task events.

## Discussion

We have shown here how cortical connectivity of intrinsic neural oscillations is regulated in adaptation to a listening challenge. These dynamics occur as spatially organized modulations in power-envelope correlations of alpha and low-beta oscillations during approximately 2-s intervals most critical for listening behavior relative to resting-state baseline. First, left fronto-parietal low-beta connectivity increased during anticipation and processing of the spatial-attention cue before speech presentation. Second, posterior alpha connectivity decreased during comprehension of speech, particularly in the interval around target-word presentation. Importantly, in subsets of participants, these connectivity dynamics predicted distinct aspects of listening behavior, namely response speed and accuracy, respectively.

### Frontoparietal β connectivity supports flexible adaptation to attentive listening

The listening task required the individuals to process the spatial cue in order to update the information about the relevance of each sentence on a trial-by-trial basis. Behaviorally, this can be viewed as an adaptation to the current listening state. Neurocognitively, this requires top-down allocation of resources responsible for attentional and task-set control.

In our representative sample, listeners with higher frontoparietal $\beta_1$ power during processing of the spatial cue showed slower word identification. Furthermore, among those listeners with resting-state frontoparietal $\beta_1$ connectivity lower than average, connectivity of the same network during processing of the spatial cue was negatively correlated with response speed in the ensuing final-word identification. Taken together, stronger frontoparietal $\beta_1$ synchrony during spatial-cue processing was associated with slower responses. Accordingly, the strength of frontoparietal $\beta_1$ connectivity likely reflects the neural cost that listeners incurred by flexibly updating and restoring the relevance of each left and right sentence on a trial-by-trial basis.

These results extend and help functionally specify the growing consensus that beta synchrony has a role in top-down control of goal-directed behaviors [51–53]. Evidence comes from animal and human studies that demonstrate a functional role of prefrontal beta synchronization in task-state transitions when content information—such as stimulus category, memory item, decision choice, or internalized task rule—need to be endogenously maintained or updated [54–57]. Task states as such have been found to coincide with short-lived beta synchrony in frontoparietal areas, e.g., during stimulus categorization [58,59], working memory retention [60,61], or accumulation of sensory evidence [62,63].

Interestingly, these findings have been reported mainly in the lower-beta frequency range (approximately 20 Hz) and often in the absence of stimulation, e.g., during delay periods of memory tasks [64]. It is generally recognized that synchrony of low-beta oscillations is initiated in higher-level control areas and propagates to lower-level sensory areas [65,66]. In addition, low-beta oscillations seem to have the optimal neurocomputational properties to prepare and maintain neural assemblies in the absence of external input [67,68]. Accordingly, momentary frontoparietal low-beta synchrony has been proposed to represent higher-level abstract task content and goals, which "awakes" an endogenous cognitive set to control a range of goal-

directed behaviors [53]. This proposal is in line with the view that frontoparietal neurons are multifunctional and distributed across multiple ensembles, which can be selectively activated by means of synchronous oscillations to flexibly enable adaptive behaviors [69,70].

Overall, our finding indicates that left-hemispheric frontoparietal $\beta_1$ connectivity acts as a top-down network mechanism to reactivate otherwise silent beta oscillations responsible for preparing cortical circuits involved in attentive listening. The strength of this connectivity may be a neural proxy of a listener's flexibility in using spatial cues to cope with challenging listening situations.

### Lower posterior cortical α connectivity supports attention to speech

When participants listened to concurrent sentences and particularly to the task-relevant final word, power-envelope correlation between posterior alpha oscillations was diminished. This finding very likely indicates a cohesive shutdown of the posterior alpha-tuned network to promote the spread of cortical information, thereby facilitating selection and comprehension of speech. We note that, during spatial cue presentation, this hypoconnectivity was only weakly observable, suggesting that posterior alpha connectivity is top-down down-regulated mainly during overt listening behavior.

Over the last 20 years, alpha oscillations have been increasingly recognized as a signature of cortical inhibition [17,71,72]. More precisely, one general view is that alpha oscillations prioritize stimulus processing by inhibiting task-irrelevant and disinhibiting task-relevant cortical areas. More recently, this view has been modified and expanded in a more mechanistic way [18]. Specifically, it has been suggested that during periods of low alpha-power cortical excitability is sufficiently high to allow continuous processing regardless of alpha phase ("medium-to-high" attentional state). In contrast, when alpha power is high, cortical processing is discontinuous and depends on the phase of alpha rhythm ("rhythmic" attentional state) [73,74]. These mechanisms appear across sensory and motor cortices [20,41,75] and point to the critical role of both power and phase dynamics of alpha oscillations in attentional control [23,73].

For alpha dynamics as such to be under top-down control, a link with brain networks must exist. Indeed, it has been previously suggested that three large-scale cortical networks differentially top-down regulate alpha oscillations [76]. Specifically, while phase synchrony of alpha oscillations has been associated with the frontoparietal network involved in adaptive control [77–79], amplitude modulation of alpha oscillations has been proposed to be under control of cingulo-opercular and dorsal attention networks involved in maintaining tonic alertness and guiding selective attention, respectively [80–82].

Overall, hypoconnectivity of posterior alpha oscillations is likely a manifestation of a broader attentional network process acting on excitability or engagement of sensory channels through their coherent release of inhibition. Our finding thus extends the previously proposed inhibitory role of alpha oscillatory activity in attention and suggests that individual attentive listening behavior hinges on both intrinsic posterior alpha amplitude-coupling and its down-regulation during selection and comprehension of speech.

### Network dynamics of the attentive listening brain: Current state and future directions

Our previous fMRI experiment [5] and the present study take a network neuroscience approach to unravel cortical underpinning of successful adaptation to a listening challenge. The fMRI study suggests that this adaptation is supported by topological reconfiguration of auditory, ventral attention, and cingulo-opercular modules. Our main finding was that functional segregation of this auditory-control network relative to its resting-state baseline

predicted individuals' listening behavior. In the present study, we found that this adaptation is supported by modulation of connectivity between alpha and low-beta neural oscillations during intervals most critical to listening behavior. Our key finding is that connectivity between frontoparietal low-beta and posterior alpha oscillations relative to their resting-state baseline allow predicting distinct aspects of individuals' listening behavior. To our knowledge, empirical evidence in support of these large-scale neural coupling dynamics and their functional relevance to adaptive behavior has not been provided yet. Thus, both studies share a common thread: Dynamics of large-scale cortical networks retain the information that could predict trait-like individual differences in attentive listening. In this way, our study also offers a novel yet feasible network identification approach, which can be used to investigate where, when, and how large-scale neural dynamics underlie adaptive behaviors.

We also realize one characteristic difference between cortical networks built upon hemodynamic signals as compared to those derived from source-reconstructed narrow-band EEG signals. In the latter case, frequency-specific cortical networks did not exhibit modular organization (S3 Fig). We additionally investigated whether measures of network centrality, namely degree centrality and eigenvector centrality, were modulated during processing of the spatial cue or final word relative to resting state. In contrast to nodal connectivity, neither of these topological measures showed significant modulations during the listening task (S11 Fig). Taken together, our data suggest that source-EEG power-envelope correlations within alpha and low-beta frequency range have an intrinsic structure whose connectivity strength (but not topology) is modulated by task context.

We note that cortical activities are organized across distributed neural processing streams and frequency channels, which are superimposed and low-pass-filtered when imaged by fMRI [83–85]. Accordingly, frequency-resolved connectivity estimation would map spectrally and spatially distinct networks depending on the neurophysiological imaging technique, e.g., electrocorticography [86–88] or M/EEG [36,89,90]. When estimation of connectivity as such is solely based on power-envelope correlations between alpha/beta oscillations in source-reconstructed M/EEG, the resulting network would be restricted to fewer cortical regions and functional connections [6,8,25,91]. Moreover, the spatial resolution with which these regions can be distinguished from one another is limited by the M/EEG sensor configuration [92]. Indeed, in our data (source-reconstructed 64-channel EEG) anterior and medial-frontal cortical regions showed sparse connectivity (Fig 4). Thus, studying topological organization of large-scale neurophysiological networks would require investigation of neural source activity over a broad frequency range with higher spatial resolution than EEG (see [93] for a recent evidence).

Attentional modulation of neural responses within auditory cortex has been extensively investigated along different lines of research in auditory neuroscience (cf. [94] for a study on auditory alpha power and neural tracking of speech using the same data and cohort as here). Nevertheless, large-scale network dynamics of the listening brain has remained underexplored, particularly on the neurophysiological level. This is mainly due to the methodological challenges inherent to connectivity analysis [95–97]. Here, we carefully took these considerations into account. This eventually allowed us to map and functionally characterize large-scale neural connectivity within alpha and low-beta frequency channels whereby distant cortical nodes tuned into attentive listening according to the listener's goal. Importantly, our brain-behavior findings further the hitherto limited understanding of behavioral differences in how individuals cope with difficult listening situations. Nonetheless, these results should be interpreted carefully as they illustrate the behavioral relevance of cortical connectivity dynamics on a between-subjects trait-like level and in subsets of participants only, depending on individual's resting-state baseline connectivity and its modulation during listening. Future studies will be

important to investigate the state-dependency of cortical connectivity dynamics, in particular their trial-by-trial variability within individuals, to further pinpoint their link to listening behavior. Looking ahead, this study opens new opportunities for brain network-based assessment of the hearing impaired as well as design of neurocognitive training strategies or assistive devices to rehabilitate or aid hearing.

## Conclusions

In sum, the present study suggests that successful adaptation to a listening challenge latches onto two distinct yet complementary neural systems: beta-tuned frontoparietal network enabling the adaptation to the attentive listening state per se, and alpha-tuned posterior cortical network supporting attention to speech. Critically, connectivity dynamics of both networks appear under top-down control, and they predict individual differences in listening behavior. To conclude, we suggest that large-scale connectivity dynamics of intrinsic alpha and low-beta neural oscillations are closely linked to the control of auditory attention.

## Materials and methods

### Data collection

This experiment was conducted as part of an ongoing large-scale study on the neural and cognitive mechanisms supporting adaptive listening behavior in healthy middle-aged and older adults ("The listening challenge: How ageing brains adapt (AUDADAPT)" https://cordis.europa.eu/project/id/646696). This project encompasses the collection of different demographic, behavioral, and neurophysiological measures across 2 time points. The analyses carried out on the data aim at relating adaptive listening behavior to changes in different facets of neural dynamics [5,94] (see also https://osf.io/28r57/).

### Participants and procedure

Reported results are from $N$ = 154 participants (right-handed German native speakers, age range = 39 to 80 y, median age = 61 y, 62 males). All participants had normal or corrected-to-normal vision, did not report any neurological, psychiatric, or other disorders, and were screened for mild cognitive impairment using the German version of the 6-Item Cognitive Impairment Test (6CIT; [98]). During EEG sessions, participants first completed 5-min eyes-open and 5-min eyes-closed resting-state measurements. The participants were asked not to think about anything specific and to avoid movement during the measurements. Next, following task instruction, participants performed 6 blocks of a demanding dichotic listening task (Fig 1). They were probed on the sentence-final noun in one of the two competing five-word sentences. As part of our large-scale study, prior to the EEG session, participants also underwent a session consisting of a general screening procedure, detailed audiometric measurements, and a battery of cognitive tests and personality profiling (see [11] for details). Only participants with normal hearing or age-adequate mild-to-moderate hearing loss were included in the present study. As part of this screening procedure, an additional 17 participants were excluded prior to EEG recording due to non-age-related hearing loss or a medical history. Three participants dropped out of the study prior to EEG recording, and an additional 10 participants were excluded from analyses after EEG recording: 3 due to incidental findings after structural MR acquisition, 6 due to technical problems during EEG recording or overall poor EEG data quality, and 1 with four task-blocks only. Participants gave written informed consent and received financial compensation (8€ per hour). Procedures were approved by the

ethics committee of the University of Lübeck (identification no. 16–107) and were in accordance with the Declaration of Helsinki.

## Dichotic listening task

All sentences had the same structure and had an average length of 2,512 ms (range: 2,183 to 2,963 ms). Root mean square intensity (−26 dB full scale (FS)) was equalized across all individual sentences, and they were masked by continuous speech-shaped noise at a signal-to-noise ratio of 0 dB. Noise onset was presented with a 50-ms linear onset ramp and preceded sentence onset by 200 ms. Each sentence pair was temporally aligned by the onset of the two task-related sentence-final nouns, which led to slightly asynchronous sentence onsets. All participants listened to the same 240 sentence pairs but in subject-specific randomized order. In addition, across participants, we balanced the assignment of sentences to the right and left ear, respectively. Each trial started with the presentation of a fixation cross in the middle of the screen (jittered duration: mean 1.5 s, range 0.5 to 3.5 s; Fig 1). Next, a blank screen was shown for 500 ms followed by the presentation of the spatial cue in the form of a circle segmented equally into two lateral halves. In selective-attention trials, one half was black, indicating the to-be-attended side, while the other half was white, indicating the to-be-ignored side. In divided-attention trials, both halves appeared in gray. After a blank screen of 500 ms duration, the semantic cue was presented in the form of a single word that specified the semantic category of both final words. The semantic category could either be given at a general (natural versus man-made) or specific level (e.g., instruments, fruits, furniture) and thus provided different degrees of semantic predictability. Each cue was presented for 1,000 ms. After a 500-ms blank-screen period, the two sentences were presented dichotically along with a fixation cross displayed in the middle of the screen. Finally, after a jittered retention period, a visual response array appeared on the left or right side of the screen, presenting 4 word-choices. The location of the response array indicated which ear (left or right) was probed. Participants were instructed to select the final word presented on the to-be-attended side using the touch screen. Among the four alternatives were the two actually presented nouns as well as two distractor nouns from the same cued semantic category. Note that because the semantic cue applied to all four alternative verbs, it could not be used to post hoc infer the to-be-attended final word. Stimulus presentation was controlled by PsychoPy. The visual scene was displayed using a 24″ touch screen (ViewSonic TD2420) positioned within an arm's length. Auditory stimulation was delivered using in-ear headphones (EARTONE 3A) at sampling rate of 44.1 kHz. Following instructions, participants performed a few practice trials to familiarize themselves with the listening task. To account for differences in hearing acuity within our group of participants, individual hearing thresholds for a 500-ms fragment of the dichotic stimuli were measured using the method of limits. All stimuli were presented 50 dB above the individual sensation level. During the experiment, each participant completed 60 trials per cue–cue condition, resulting in 240 trials in total. The cue conditions were equally distributed across 6 blocks of 40 trials each (approximately 10 min) and were presented in random order. Participants took short breaks between blocks.

## EEG data analysis

**Data acquisition.** Participants were seated comfortably in a dimly lit, sound-attenuated recording booth where we recorded their EEG from 64 active electrodes mounted to an elastic cap (Ag/AgCl; ActiCap/ActiChamp, Brain Products, Gilching, Germany). Electrode impedances were kept below 30 kΩ. The signals were digitized at a sampling rate of 1,000 Hz and referenced online to the left mastoid electrode (TP9, ground: AFz). For subsequent offline EEG data analyses, we used the EEGlab [99] (version 14-1-1b) and Fieldtrip [100] (version 2016-06-13) toolboxes for Matlab and other customized scripts.

**Artifact rejection.** First, rest and task continuous data were concatenated, band-pass filtered (1 to 100 Hz), down-sampled (300 Hz), and cut into 1-s epochs. The resulting time-domain data were then subjected to visual inspection by trained assistants to discard nonstereotypical epochs (i.e., short-lived high-variance artifacts) and to identify and remove noisy channels. Subsequently, the data were re-referenced to the average of all EEG channels (average reference). Next, using EEGlab's default runica implementation of independent component analysis (ICA), the data were decomposed into 63 components (or less if noisy channels were removed). The results were then carefully inspected by trained assistants to classify non-brain components including eye blinks and lateral eye movements, muscle activity, heart beats, and single-channel noise. This classification was based on components' topographies, time courses, spectra, and source dipole locations in EEGlab. Finally, only brain components were back-projected separately per rest and task data, and noisy channels (if removed previously) were interpolated using the nearest neighbor approach.

**Preprocessing.** The artifact-clean continuous EEG data were high-pass-filtered at 0.3 Hz (finite impulse response (FIR) filter, zero-phase lag, order 5,574, Hann window) and low-pass-filtered at 180 Hz (FIR filter, zero-phase lag, order 100, Hamming window). Task data were cut into 10-s trials within −2 to 8 s relative to the onset of the spatial-attention cue to capture cue presentation as well as the entire auditory stimulation interval. Resting-state data were similarly cut into 10-s epochs. Data were downsampled to fs = 250 Hz. These procedures were implemented in Fieldtrip. Trials during which the amplitude of any individual channel exceeded a range of 200 microvolts were removed before main analyses.

**Time-frequency analysis.** Spectro-temporal estimates of single-trial task data were obtained for a time window of −1 to 7.5 s (relative to the onset of the spatial-attention cue) at 50-ms resolution and frequencies ranging from 2 to 32 Hz on a logarithmic scale (Morlet wavelets; number of cycles = 6, Fieldtrip implementation). The results were used for analyzing power modulation of brain oscillatory responses on a sensor and source level (S7 Fig). As a measure of nodal power, we used the mean of trial-average baseline-corrected source power within each time-frequency window of interest. The same time-frequency analysis was used for estimating power-envelope correlations during rest and task (see below).

**EEG source and forward model construction.** Thirty participants had completed functional and structural magnetic resonance imaging (MRI) performing the same experiment on a separate session. For these participants, individual EEG source and forward models were created based on each participant's T1-weighted MRI image. The T1 image of 1 female and 1 male participant was used as a template for the remaining participants. First, anatomical images were resliced (256 × 256 × 256 voxels, 1-mm resolution), and the CTF convention was imposed as their coordinate system. Next, cortical surface of individual (or template) T1 image was constructed using Freesurfer function recons-all (S1 Fig). The result was used to generate a cortical mesh in accordance with Human Connectome Project (HCP) standard atlas template available under https://github.com/Washington-University/HCPpipelines by means of HCP Workbench (version 1.5; wb_command using Fieldtrip script ft_postfreesurferscript.sh). This gave individual cortical mesh encompassing 4,000 grid points per hemisphere, which was then used as the individual EEG source model geometry. This ensured that we have the same number of grid points per individual cortical mesh and that the location of these points is comparable across participants. This is particularly relevant when adopting the HCP functional parcellation template during source projection of sensor data as described in Power-envelope correlation analysis section. To generate individual head and forward model, first T1 images were segmented into 3 tissue types (brain, skull, and scalp) using FieldTrip function ft_volumesegment (S1 Fig). Subsequently, the forward model (volume conduction) was estimated using boundary element method in Fieldtrip ("dipoli" implementation). Next, the fit of

digitized EEG channel locations (xensor digitizer, ANT Neuro) to individuals' head models were optimized in CTF coordinate system using rigid-body transformation and additional interactive alignment. Finally, the lead-field matrix at each channel × source location was computed using the source and forward model per individual (S1 Fig).

**Beamforming.**　Source reconstruction (inverse solution) was achieved using a frequency-domain beamforming approach, namely partial canonical coherence (PCC) [95]. To this end, first 5-min eyes-open resting state and a random 5-min split of task data were concatneted. This procedure prevented a potential bias in estimation of the spatial filter due to difference in the duration of data between rest and task. Next, cross-spectral density of the data was calculated using a broad-band filter centered at 15 Hz (bandwidth = 14 Hz). The result together with the source and forward model (see above) was used in Fieldtrip to obtain a common spatial filter per individual (regularization parameter: 5%, dipole orientation: axis of most variance using singular value decomposition; see S1 Fig).

**Source projection of sensor data.**　Sensor-level single-trial (epoch) complex-value time-frequency estimates were projected to source space by matrix-multiplication of the common spatial filter weights (S1 Fig). To increase signal-to-noise ratio, reduce redundancy in source-projected data, and computationally facilitate connectivity analyses, individual source-projected data were averaged across cortical surface grid-points per cortical patch defined according to the HCP functional parcellation template ([46]; similar to [37]). This parcellation provides a symmetrical delineation of each hemisphere into 180 parcels. This gave single-trial (or single-epoch) time-frequency source estimates at each cortical node. We note that a cortical parcellation with 360 nodes over-sample the realistic spatial resolution of 64-channel EEG. However, an optimal EEG-based cortical parcellation is currently not available, although studies have begun to fill this gap [92]. Therefore, we used an established fMRI-based parcellation to approximate spatial boundaries of frequency-specific EEG source connectivity and its modulation throughout the listening task. This also allowed us to investigate the similarities and differences between fMRI and EEG whole-brain connectivity during rest and the same listening task in a subgroup of participants.

## Connectivity analysis

Frequency-resolved connectivity analysis was done in the following steps per individual (see S1 Fig). First, single-trial time-frequency source estimates were concatenated across time per frequency. For estimating resting-state connectivity, this was done based on 5-min epoched data. For estimating overall connectivity during listening task irrespective of trial intervals (Fig 4), this was done by selecting 30 random trials per task block (equivalently, 5-min data as in rest). Then, connectivity matrices were averaged across 6 task blocks. This procedure assured that rest and task connectivity did not artificially differ due to differences in the duration of data used for estimating connectivity [101]. For estimating task connectivity during each 1-s time window (e.g., spatial-cue or final-word period; Fig 1, colored intervals), windowed signals were concatenated across all 240 trials (equivalently, 4-min data). Event-related task connectivity obtained from this procedure was then compared with 4-min resting state connectivity. To investigate the effect of block on connectivity (see Statistical analysis below), the same procedure was repeated based on all 40 trials per block. For this latter analysis, resting state connectivity was estimated based on 40 s of data. Thus, in all analyses, data were matched in duration across rest, task, and all participants before estimation of connectivity.

**Power-envelope correlation analysis.**　To assess frequency-specific neural interactions, we computed Pearson's correlations between the log-transformed power of all pairs of nodes (all-to-all connectivity). To eliminate the trivial common covariation in power measured from

the same sources, we used the orthogonalization approach proposed by [25] prior to computing the power correlations (Fieldtrip implementation). This approach has been suggested and used to circumvent overestimation of instantaneous short-distance correlations, which can otherwise occur due to field propagation [36,50,102]. The above procedure yielded frequency-specific 360-by-360 functional connectivity matrices per participant and carrier frequency (and trial interval). The between-subject reliability analysis (see S2 Fig for details) suggested that power-envelope correlations were strongest and showed highest reliability within 7 to 24 Hz. Within this range, 3 frequency bands were defined. Connectivity matrices in each frequency band was derived by averaging corresponding frequency-specific connectivity matrices. Change in connectivity was evaluated by first calculating task-connectivity matrix minus rest-connectivity matrix per frequency, and then averaging the results within each frequency band. Finally, connectivity matrices were thresholded at 10% of network density (proportional thresholding) [101]. This procedure ensured that networks were matched in density across rest, task, and all participants. Subsequently, nodal connectivity was obtained by calculating the sum of each node's connection weights (i.e., correlations values). Mean connectivity of a network (e.g., the frontoparietal network) was obtained by averaging nodal connectivity values across the network.

## Statistical analysis

**Behavioral data.** Participants' behavioral performance in the listening task was evaluated with respect to accuracy and response speed. Trials in which participants failed to answer within the given 4-s response window ("timeouts") were excluded from the analysis. Spatial stream confusions, i.e., trials in which the final word of the to-be-ignored speech stream were selected, and random errors were jointly classified as incorrect answers. The analysis of response speed, defined as the inverse of reaction time, was based on correct trials only.

**Connectivity data.** Statistical comparisons of nodal and mean connectivity between rest and task were based on permutation tests for paired samples (randomly permuting the rest and task labels 10,000 times). We used Cohen's d for paired samples as the corresponding effect size. For nodal connectivity analysis, and to correct for multiple comparisons entailed by the number of cortical nodes, we used FDR procedure at significance level of 0.01 (two-sided). Knowing the skewed distribution of mean connectivity, these valuse were logit-transformed before submitting to (generalized) linear mixed-effects models (see below).

**Brain-behavior models.** Brain-behavior relationship was investigated within a linear mixed-effects analysis framework. To this end, either of the single-trial behavioral measures (accuracy or response speed) across all participants were treated as the dependent variable. The main experimental predictors in the model were the single-trial spatial and semantic cue conditions, each at 2 levels (divided versus selective and general versus specific, respectively). Neural predictors entered as between-subject regressors. These include mean resting state connectivity, mean event-related task connectivity per block, and mean event-related neural oscillatory power (dB) per block. The linear mixed-effects analysis framework allowed us to account for other variables, which entered as regressors of no-interest in the model. These include age, mean pure-tone audiometry (PTA) averaged across left and right ear, side probed (left or right), and task-block number. Mixed-effects analyses were implemented in R using the packages lme4 [103], effects [104], and sjPlot (https://strengejacke.github.io/sjPlot/).

**Model estimation.** The regressors in each brain-behavior model included the main effects of all predictors introduced above, plus the interaction between the two listening cues and the interaction between rest and task connectivity. Additional interaction terms were also included to explore their possible effects on listening behavior. The influence of listening cues

and of neural measures were tested in same brain-behavior model. The models also included random intercepts by subject. In a data-driven manner, we then tested whether model fit (performed using maximum-likelihood estimation) could be further improved by the inclusion of subject-specific random slopes for the effects of the spatial-attention cue, semantic cue, or probed ear. The change in model fit was assessed using likelihood ratio tests on nested models. Deviation coding was used for categorical predictors. All continuous variables were z-scored. For the dependent measure of accuracy, we used generalized linear mixed-effects model (binomial distribution, logit link function). For response speed, we used general linear mixed-effects model (gaussian distribution, identity link function). Given the large sample size, p-values for individual model terms are based on Wald t-as z-values for linear models [105] and on z-values and asymptotic Wald tests in generalized linear models. All reported p-values are corrected to control for the FDR. As a measure of effects size, for the model predicting accuracy, we report ORs, and for response speed, we report the regression coefficient (β).

**Bayes factor.** To facilitate the interpretability of significant and nonsignificant effects, we calculated the BF based on the comparison of Bayesian information criterion (BIC) values as proposed in [106]: BF = exp([BIC(H0)−BIC(H1)]/2). To calculate the BF for a given term, we compared the BIC values of the full model to that of a reduced model in which only the rest-connectivity × task-connectivity interaction term was removed. By convention, log-BFs larger than 1 provide evidence for the presence of an effect (i.e., the observed data are more likely under the more complex model), whereas log-BFs smaller than −1 provide evidence for the absence of an effect (i.e., the observed data are more likely under the simpler model) [47].

## Data visualization

Brain surfaces were visualized using the Connectome Workbench. Brain-behavior interaction plots were visualized using R package effects [104].

## Supporting information

**S1 Fig. Hypothetical and methodological framework. (A)** Toy graphs illustrate connectivity between 3 exemplary EEG cortical sources as measured by power-envelope correlation. During attentive listening, intrinsic α/β neural oscillations reconfigure their putative baseline network (resting state) depending on the current task state to support listening behavior. These dynamics could manifest as change in connectivity strength (here depicted as edge thickness) or network segregation [5]. **(B)** EEG source analysis pipeline. Source reconstruction of EEG oscillatory responses and estimation of connectivity per individual ($N = 154$) were implemented in four steps: (1) construction of source geometry using MRI T1-derived cortical mesh and head boundary-element model (2) estimation of lead-field matrix and a spatial filter common across rest, task, and a broad frequency band (fc = 15 Hz; BW = 14 Hz) (3) source projection of time-frequency sensor data and averaging the source estimates per cortical parcel or node (4) pair-wise orthogonalization of complex-value time-frequency source estimates across all nodes to diminish spurious correlations due to volume conduction [25]. These steps at the end gave individual cortical connectivity maps per time-window of interest and frequency band. These analyses were performed using Fieldtrip toolbox for Matlab and HCP functional parcellation template [46]. BW, band-width; EEG, electroencephalography; fc, center frequency; fs, sampling frequency; HCP, Human Connectome Project; ICA, independent component analysis; MRI, magnetic resonance imaging; PCC, partial canonical coherence.
(TIF)

**S2 Fig. Between-subject reliability of power-envelope correlation.** A serious but often neglected potential confounding factor in assessments of neuronal interactions is SNR [48,49]. This is an important issue in the present study as we compare connectivity of frequency-specific neural oscillations between resting state and listening task, each of which is potentially measured at different levels of SNR. Thus, we first investigated at which frequencies connectivity can be reliably measured under both rest and task conditions. As a measure of reliability, we used between-subject correlation of rest and task nodal connectivity values, i.e., columns of raw connectivity matrices. The reliability analysis was done in two steps. **(A, B)** First, per cortical node, correlations between nodal connectivity values were calculated across all pairs of $N = 154$ participants, separately for rest or task condition. This procedure per node gives one symmetric $N \times N$ correlation matrix for each rest or task condition. The upper-diagonal average of this matrix is mean between-subject correlation in nodal connectivity for each rest or task condition (within-condition reliability). The plots in (A) and (B) illustrate mean ± SEM of these between-subject correlations averaged across all 360 cortical nodes per frequency. **(C)** Next, the same analysis was done across rest and task. In this case, the result per node is one asymmetric $N \times N$ correlation matrix. The off-diagonal average of this matrix is mean between-subject correlation in nodal connectivity across rest and task (between-condition reliability). The black line graph in (C) illustrates mean ± SEM of these between-subject correlations averaged across all 360 cortical nodes per frequency. These correlations are underestimated in the presence of noise. Thus, they were submitted to Spearman's correction for attenuation to account for differences in SNR between rest and task (blue line graph). Estimation of connectivity at a given frequency and node was considered reliable if the attenuation-corrected correlation was consistently positive across all participants. Plots in second row illustrate percentage of cortical nodes showing significantly positive between-subject correlation ($p < 0.05$, uncorrected). As illustrated by both blue line graphs, between-subject correlations were consistently positive across all nodes within the frequency range 7–32 Hz ($0.1 < r < 0.4$). These results illustrate that power-envelope correlations between EEG oscillatory sources can be reliably measured within alpha and low-beta frequency range (gray frequency intervals). The data underlying this figure can be found at https://osf.io/ge2cq/. SNR, signal-to-noise ratio.
(TIF)

**S3 Fig. Network modularity of α/β oscillations under rest and attentive listening.** In each group-average connectivity matrix, diagonal squares represent functional modules. Off-diagonal correlations represent the strength of between-module connections. In (A), functional modules correspond to canonical resting state networks (left) and their makeup during the listening task (right; see [5]). In contrast, EEG source power-envelope correlations within alpha and low-beta bands did not exhibit a modular organization. Connectivity matrices are averaged across individuals and thresholded at 10% of network density. Modularity index ($Q$) quantifies the degree to which a network is clustered into densely intra-connected groups of nodes, which are sparsely inter-connected and is estimated based on Newman optimization algorithm [44,45] (the more modular the network, the closer the $Q$ to 1). Percentage of participating nodes is calculated as the number of nodes having at least one connection in the network divided by the total number of nodes defined according to the cortical parcellation template [46]. The data underlying this figure can be found at https://osf.io/ge2cq/. EEG, electroencephalography; fMRI, functional magnetic resonance imaging.
(TIF)

**S4 Fig. Frequency-specific connectivity matrices. (A)** For each frequency band, power-envelope correlations between EEG oscillatory sources were estimated using 4-min eyes-open

resting state data. **(B)** Whole-brain connectivity during task time intervals most critical to listening behavior. Power-envelope correlations were estimated by concatenating 1-s windowed signals across all 240 trials (4-min data). Note the stronger beta-band connectivity during processing of spatial cue and weaker alpha-band connectivity during final word presentation in (B) relative to resting state (A). Connectivity maps were averaged across $N = 154$ individuals and thresholded at 10% of network density. Nodes correspond to cortical parcels as in [46] and are grouped according to their cortical lobes. The data underlying this figure can be found at https://osf.io/ge2cq/. EEG, electroencephalography; LH, left hemisphere; RH, right hemisphere.
(TIF)

**S5 Fig. Cortical connectivity dynamics of α/β oscillations during anticipation and cueing.** For each time interval (A-C) and frequency band, power-envelope correlations between EEG oscillatory sources were estimated by concatenating 1-s windowed signals across all 240 trials (4-min data) and compared with 4-min resting state connectivity at the same frequency band (i.e., task minus rest). In anticipation of and during the spatial cue presentation, $\beta_1$ connectivity was significantly increased mainly across frontoparietal regions (A and B, third column; paired-sample permutation tests; significant nodes are outlined in black). Alpha-band connectivity was not significantly different than rest. Connectivity difference maps were averaged across $N = 154$ individuals and thresholded at 10% of network density. Nodes correspond to cortical parcels as in [46] and are grouped according to their cortical lobes. The data underlying this figure can be found at https://osf.io/ge2cq/. EEG, electroencephalography; LH, left hemisphere; RH, right hemisphere.
(TIF)

**S6 Fig. Cortical connectivity dynamics of α/β oscillations during sentence presentation.** For each time interval of sentence presentation (A-C) and frequency band, power-envelope correlations between EEG oscillatory sources were estimated by concatenating 1-s windowed signals across all 240 trials (4-min data) and compared with 4-min resting state connectivity at the same frequency band (i.e., task minus rest). Toward the end of sentence and particularly during final word presentation, alpha-band connectivity was significantly decreased across posterior cortical regions (A-C, first column; paired-sample permutation tests; significant nodes are outlined in black). Beta-band connectivity was not significantly different than rest during sentence presentation. Connectivity difference maps were averaged across $N = 154$ individuals and thresholded at 10% of network density. Nodes correspond to cortical parcels as in [46] and are grouped according to their cortical lobes. The data underlying this figure can be found at https://osf.io/ge2cq/. EEG, electroencephalography; LH, left hemisphere; RH: right hemisphere.
(TIF)

**S7 Fig. Modulation of neural oscillatory activity during attentive listening. (A)** Time-frequency representation in EEG sensor space. Power estimates were averaged across all sensors, trials, and $N = 154$ participants. **(B)** The same as in (A) but based on power of EEG source estimates and averaged across all cortical nodes. Power changes are calculated as dB change relative to the whole-trial baseline interval ($-1$–7.5 s). **(C)** Attentional modulation of alpha source power during selective attention to final word. The modulation index is calculated as AMI = ($\alpha$-power$_{attendL}$ − $\alpha$-power$_{attendR}$) / ($\alpha$-power$_{attendL}$ + $\alpha$-power$_{attendR}$). The data underlying this figure can be found at https://osf.io/ge2cq/. AMI, attentional modulation index; EEG, electroencephalography; LH, left hemisphere; RH, right hemisphere.
(TIF)

**S8 Fig. Correlation between neural activity power and connectivity during intervals of listening task. (A)** For each frequency band and time interval, correlation between nodal connectivity and mean nodal power (dB) was tested across all $N = 154$ participants and per cortical node. **(B)** The same analysis as in (A) but using measures averaged across frontoparietal or posterior cortical nodes involved in beta-band hyperconnectivity or alpha-band hypoconnectivity, respectively. The data underlying this figure can be found at https://osf.io/ge2cq/. n.s., not significant.
(TIF)

**S9 Fig. EEG source connectivity was not influenced by spatial cue conditions.** We investigated whether alpha- or beta-band connectivity showed a similar hemispheric lateralization in response to the spatial cue as in alpha power. For each spatial-cue (A, B) or final-word (C, D) time interval and frequency band, power-envelope correlations between EEG oscillatory sources were estimated by concatenating 1-s windowed signals across trials per spatial-cue condition: selective attention (i.e., attend left or right), divided attention, attend left, or attend right. The results were then compared by first calculating connectivity difference maps per participant (e.g., selective minus divided) and then averaging the difference maps across $N = 154$ individuals. If connectivity is confounded by activity level, these maps should reveal differences in power-envelope correlations between conditions. In contrast, power-envelope correlations were not influenced by attentional-cue conditions in neither of the trial intervals nor frequency bands. None of the connectivity contrasts revealed significant differences in nodal connectivity when tested across all cortical nodes using paired-sample permutation tests. Nodes correspond to cortical parcels as in [46] and are grouped according to their cortical lobes. The data underlying this figure can be found at https://osf.io/ge2cq/. EEG, electroencephalography; LH, left hemisphere; RH, right hemisphere.
(TIF)

**S10 Fig. Within-sample reproducibility and likelihood assessment of the brain-behavior interaction effects.** The reliability and robustness of the brain-behavior interaction effects (main text, Fig 6) were investigated by randomly splitting the data into k = 5 nonoverlapping folds. The strength and uncertainty (i.e., confidence interval) of the model's parameter estimate was then examined when each fold of data was used. Since this analysis does not take the number of observations and model complexity into account, it could be that the model has a relatively high goodness of fit as a too complex model is fitted to the data. We thus examined the relative strength of evidence (rather than statistical significance) in support of the alternative hypothesis (i.e., model with the interaction term) as compared to the null hypothesis (i.e., model without the interaction term). This was done by calculating the BIC approximation of the BF, i.e., exp([BIC(H0)−BIC(H1)]/2). In this calculation, BIC imposes penalty on each model log-likelihood based on both number of observations and parameters. Following Harrold Jeffery's scale for interpretation of BF, $0 < \log\text{-BF} < 1$ suggests weak evidence for H1, $1 < \log\text{-BF} < 3$ is positive, and $3 < \log\text{-BF} < 5$ is strong [104]. $\beta$: Slope parameter estimates from linear mixed-effects model. OR: odds ratio parameter estimates from generalized linear mixed-effects models. The data underlying this figure can be found at https://osf.io/ge2cq/. BF, Bayes factor; BIC, Bayesian information criterion; OR, odds ratio.
(TIF)

**S11 Fig. Network topology and centrality of α/β oscillations under rest and attentive listening. (A)** Network topology per frequency band and condition. Each connectogram corresponds to its connectivity matrix counterpart and is obtained by setting the connection weights to 1 for correlations higher than 95th percentile, and to 0 otherwise (i.e., a sparse

binary graph thresholded at 5% of density). Nodes correspond to cortical parcels as in [46] and are grouped according to their cortical lobes. The frontoparietal and posterior cortical nodes showing significant modulation of $\beta_1$ and $\alpha_1$ connectivity strength, respectively (main text, Fig 5), are depicted in black. **(B)** Comparison of nodal degree centrality between each task interval and rest using permutation tests. Degree centrality is defined as each node's number of connections (also referred to as node's degree [44]). This measure quantifies relative importance of nodes: Nodes having large number of connections, hence high degree centrality, can be considered as central coordinators in the functional network. In contrast to nodal connectivity strength, nodal degree was not significantly modulated when tested across all cortical nodes (brain surfaces; significance level: $p_{FDR} < 0.01$). **(C)** Comparison of eigenvector centrality. This metric is more specific than degree centrality as it also takes the relative importance of a node's neighbors into account. Eigenvector centrality of node $i$ is equivalent to the $i$th element in the eigenvector corresponding to the largest eigenvalue of the adjacency matrix. Nodes having high eigenvector centrality are high-degree nodes whose neighbors also have relatively large number of connections. Eigenvector centrality was not significantly modulated when tested across all cortical nodes. The data underlying this figure can be found at https://osf.io/ge2cq/. dlPFC, dorsolateral prefrontal cortex; FEF, frontal eye field; IFG, inferior frontal gyrus; IPL, inferior parietal lobule; LH, left hemisphere; n.s., not significant; PSL, perisylvian language area; RH, right hemisphere; STS, superior temporal sulcus.
(TIF)

**S1 Table. Summary table of the generalized linear mixed-effects model predicting individuals' listening accuracy.** In this model, brain regressors were based on the data during final word period and within $\alpha_1$ band. Significant effects after FDR-correction for multiple comparisons across model terms are shown in blue. The main effects of and interactions between listening cues are visualized in Fig 2 (main text). The interaction between rest and task connectivity is visualized in Fig 6. OR: odds ratio; $\sigma^2$: within-group variance; $\tau_{00}$: between-group variance; $\rho_{01}$: random-slope-intercept-correlation. FDR, false discovery rate; OR, odds ratio.
(TIF)

**S2 Table. Summary table of the linear mixed-effects model predicting individuals' response speed.** In this model, brain regressors were based on the data during final word period and within $\alpha_1$ band. Significant effects after FDR-correction for multiple comparisons across model terms are shown in blue. The main effects of and interactions between listening cues are visualized in Fig 2 (main text). $\beta$: slope parameter estimate; $\sigma^2$: within-group variance; $\tau_{00}$: between-group variance; $\rho_{01}$: random-slope-intercept-correlation. FDR, false discovery rate.
(TIF)

**S3 Table. Summary table of the generalized linear mixed-effects model predicting individuals' listening accuracy.** In this model, brain regressors were based on the data during final word period and within $\alpha_2$ band. Significant effects after FDR-correction for multiple comparisons across model terms are shown in blue. OR: odds ratio; $\sigma^2$: within-group variance; $\tau_{00}$: between-group variance; $\rho_{01}$: random-slope-intercept-correlation. FDR, false discovery rate; OR, odds ratio.
(TIF)

**S4 Table. Summary table of the linear mixed-effects model predicting individuals' response speed.** In this model, brain regressors were based on the data during final word period and within $\alpha_2$ band. Significant effects after FDR-correction for multiple comparisons

across model terms are shown in blue. $\beta$: slope parameter estimate; $\sigma^2$: within-group variance; $\tau_{00}$: between-group variance; $\rho_{01}$: random-slope-intercept-correlation. FDR, false discovery rate.
(TIF)

**S5 Table. Summary table of the generalized linear mixed-effects model predicting individuals' listening accuracy.** In this model, brain regressors were based on the data during spatial cue and within $\beta_1$ band. Significant effects after FDR-correction for multiple comparisons across model terms are shown in blue. OR: odds ratio; $\sigma^2$: within-group variance; $\tau_{00}$: between-group variance; $\rho_{01}$: random-slope-intercept-correlation. FDR, false discovery rate; OR, odds ratio.
(TIF)

**S6 Table. Summary table of the linear mixed-effects model predicting individuals' response speed.** In this model, brain regressors were based on the data during spatial cue period and within $\beta_1$ band. Significant effects after FDR-correction for multiple comparisons across model terms are shown in blue. The main effects of and interactions between listening cues are visualized in Fig 2 (main text). The interaction between rest and task connectivity is visualized in Fig 6. $\beta$: slope parameter estimate; $\sigma^2$: within-group variance; $\tau_{00}$: between-group variance; $\rho_{01}$: random-slope-intercept-correlation. FDR, false discovery rate.
(TIF)

## Acknowledgments

Parts of the connectivity analyses were conducted using the OMICS compute cluster at the University of Lübeck. The authors are grateful for the help of Franziska Scharata in acquiring the data.

## Author Contributions

**Conceptualization:** Mohsen Alavash, Sarah Tune, Jonas Obleser.

**Data curation:** Mohsen Alavash, Sarah Tune.

**Formal analysis:** Mohsen Alavash.

**Funding acquisition:** Mohsen Alavash, Jonas Obleser.

**Methodology:** Mohsen Alavash, Sarah Tune, Jonas Obleser.

**Project administration:** Mohsen Alavash, Sarah Tune, Jonas Obleser.

**Resources:** Jonas Obleser.

**Supervision:** Jonas Obleser.

**Visualization:** Mohsen Alavash.

**Writing – original draft:** Mohsen Alavash.

**Writing – review & editing:** Sarah Tune, Jonas Obleser.

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
