## [Editor Report · Decision Letter 0]

3 Mar 2021

Dear Dr Alavash, 

Thank you for submitting your manuscript entitled "Large-scale cortical connectivity dynamics of intrinsic neural oscillations support adaptive listening behavior" for consideration as a Research Article by PLOS Biology.

Your manuscript has now been evaluated by the PLOS Biology editorial staff, as well as by an academic editor with relevant expertise, and I am writing to let you know that we would like to send your submission out for external peer review.

Please re-submit your manuscript within two working days, i.e. by Mar 05 2021 11:59PM.

Kind regards,

Lucas Smith, Ph.D.,

Associate Editor

PLOS Biology

---

## [Decision Letter · Decision Letter 1]

4 May 2021

Dear Dr Alavash,

Thank you very much for submitting your manuscript "Large-scale cortical connectivity dynamics of intrinsic neural oscillations support adaptive listening behavior" for consideration as a Research Article at PLOS Biology. Your manuscript has been evaluated by the PLOS Biology editors, an Academic Editor with relevant expertise, and by several independent reviewers.

The reviews are appended below. As you will see, the reviewers note that the methods used here are robust, and that the data is largely convincing. However, the reviewers have raised several concerns which will need to be addressed before we can consider your manuscript for publication. For example, Reviewer 2 questions the novelty of this study in relation to previous work on functional connectivity and attention. After discussing this feedback with the Academic Editor, we think you should be given the opportunity to clarify how your findings are different from the published record and should be published in a general biology journal. It will be important to add discussion of what is known in other systems about attention and functional connectivity, and to highlight what is added by the current work.

In light of the reviews, we will not be able to accept the current version of the manuscript, but we would welcome re-submission of a much-revised version that takes into account the reviewers' comments. We cannot make any decision about publication until we have seen the revised manuscript and your response to the reviewers' comments. Your revised manuscript is also likely to be sent for further evaluation by the reviewers.

We expect to receive your revised manuscript within 3 months. 

**IMPORTANT - SUBMITTING YOUR REVISION**

*Re-submission Checklist*

*Published Peer Review*

*PLOS Data Policy*

*Blot and Gel Data Policy*

Sincerely,

Lucas Smith

Associate Editor

PLOS Biology

lsmith@plos.org

REVIEWS:

Reviewer #1: Alavash et al. re-analysed this large dataset to address the question how EEG (power and) connectivity in alpha and beta frequency bands affect speech comprehension in a dichotic listening task. The study used an age-varying large sample of N=154 / N=127 after exclusion (I think). The authors found that in individuals with low frontoparietal beta connectivity during rest, there was a link between response speed and this connectivity during spatial cue processing. They also found that in individuals with high posterior alpha1 connectivity during rest, accuracy was influenced by this connectivity during final-word presentation.

The topic should be of interest to speech researchers and cognitive and systems neuroscientists. The methods are robust. The manuscript is largely well written and easy to follow. The authors have adopted an interesting strategy with this manuscript, because many methodological details and statistics can only be found in the Supplemental Information. This leads to a very straightforward manuscript, but also bears the risk of burying important info in SI. I would suggest a more "traditional" approach and move important info into the main manuscript (cleaning of data, source localisation, statistical main effects, even if not significant). It seems a very narrow picture to report only the most interesting results (the significant interactions) and not mention the other non-significant results. Related to this, the discussion sometimes gives the impression that strong main effects were found, and not the actually found interactions, which indicate that there were connectivity-behaviour links in a subset of participants. This should be expressed more carefully (and discussed) throughout to better reflect the statistical results. 

Please note that I used the initial submission to review this manuscript, because the format of this was more reviewer-friendly (figures in-text) and printer-friendly (fewer pages).

Other points:

1. How was the decision made to split the analysis into these specific frequency ranges? Please elaborate/justify.

2. I am not quite sure what the message in the paragraph starting in line 67 is (initial submission). Is it that alpha and beta oscillations are modulated by slow time scales <.1 Hz? Could the authors clarify?

3. Please briefly explain the use of "modular organization" (l. 183) in this context.

4. The control analysis are great, but for the cross-validation results (starting line 306), I wonder what it means if 3 (and 1, respectively) out of 5 model iterations are significant (log BF larger than 1)? This does not seem to be the most robust? Could the authors add (a little) more information on how to reasonably interpret this?

5. Many of the supplemental figures miss detailed captions. Could the authors please add these?

6. Please explicitly report the final sample size and demographic description of the final sample.

7. Related to this, I would suggest detailing the actual sample size in the abstract, instead of the initial number of participants before exclusion.

Reviewer #2: The paper complements an earlier fMRI connectivity study by the same authors in which they used resting state data and an attentional listening paradigm. The group now provides an analysis of amplitude envelope coupling in EEG data using the same task paradigm. 

While the EEG recordings use the minimum number of channels (64 electrodes) required for meaningful source analysis, an interesting feature of the study is a large sample, since data from more than 150 subjects were included. 

The results replicate known resting state envelope coupling patterns. Furthermore, the study shows that auditory attention modulates envelope coupling in the beta band, leading to higher functional connectivity of frontoparietal regions compared to the resting state. For subgroups of the subjects, connectivity values predict performance in the task. 

The state-of-the-art is not properly covered in the introduction; for instance, there is substantial work on network dynamics during crossmodal attentional shifts involving the auditory system which are not covered here. Also, the statement that studies on visuo-spatial attention have provided „initial evidence" on a role of coupling for attention (p.6, line 65) is misleading; there is substantial evidence, and in my view it would rather be fair to say that these studies have pretty much resolved the role of coupling for attentional selection.

In the task design, spatial attention and feature-based attention are mixed; the rationale for combining these two aspects is not clearly presented.

The age range of subjects included is pretty broad; it would be interesting to know whether there were significant age-related differences for resting state or task-related envelope coupling.

Overall, the study is solid and uses an interesting paradigm. However, on the background of previous work on functional connectivity and attention, novelty is limited. 

Reviewer #3: This study investigated oscillatory power correlations in elderly adults using EEG. The study found that frontoparietal beta-band connectivity was increased while posterior alpha-band connectivity was decreased during challenging dual-talker listening task. The manuscript is well written and coherent, the results being presented in a clear manner. The analyses seem also well performed and there is extensive amount of control analysis included which I found very convincing. The illustration of the results could be improved to better convey the message but overall the study is of very good quality. 

Major

In the abstract it is written that "connectivity between intrinsic neural oscillations extracted from source-reconstructed electroencephalography is top-down regulated during a challenging dual-talker listening task." However, there is no directional analysis in this study and the sentence should be reformulated to reflect this. The localization of the connectivity is suggestive of the top-down influence, but this should be separately tested. 

There is no summary of the connectivity results, which would show connectivity changes as a function of frequency for different conditions e.g. as time-frequency representations of connectivity. Thus, it is not evident that the effects would be found in the alpha and beta bands now used in the analyses. These data might be presented in Fig S2d-e but there is no legend for these, so it is unclear. I suggest that these are moved to main figures, if they are the main data of the findings. 

The lack of low-frequency oscillations to the final world could also arise if the final world is not time-locked to the beginning of the sentence and thus the oscillatory signals being averaged out. Please clarify how the timing of the very long trials with variable position of the final words were ensured? 

Connectivity data is presented as matrix correlation coefficients between different lobes as well as nodal connectivity overlaid in the brain surface. I find these figures suboptimal. The connectivity estimates presented in Figs 4, 5 are not informative of the anatomy nor strength of the connections, because the parcels are not identifiable neither from the matrixes (which should have the labels at least in some panels) nor from the brain which are really blurred and not revealing the brain areas. Maybe the authors could add some land-mark brain regions relevant to auditory attention and semantic processing. Also color scale could be more informative to reveal differences in the strength of the connectivity. 

Nodal connectivity is now used to plot the cortical regions with connectivity modulations. Strength based metrics are not the best to use as they are influenced by the source-leakage (the poor sources have the largest signal). Centrality measures are more robust in this respect and they also give some insight of the network structure and its relevance to information processing. I would thus suggest trying alternative approaches to visual the data to increase the clarity of the results. 

Minor

Morlet wavelets from 1-32 Hz were used to analyse oscillation power. Fig S7 clearly shows that there would be oscillations also above the 32Hz so why this cut-off was used? Gamma-band oscillations from 30-100 Hz are known to be related to the processing of auditory semantic information.

---

## [Decision Letter · Decision Letter 2]

16 Aug 2021

Dear Dr Alavash,

Thank you for submitting your revised Research Article entitled "Large-scale cortical connectivity dynamics of intrinsic neural oscillations support adaptive listening behavior" for publication in PLOS Biology. I have now obtained advice from two of the original reviewers and have discussed their comments with the Academic Editor. 

As you will see, both reviewers are largely satisfied with the revision, although Reviewer 1 has noted a number of points that require additional explanation and/or clarifications. Therefore, we will probably accept this manuscript for publication, provided you satisfactorily address the remaining points raised by the reviewer 1. **IMPORTANT: Please also make sure to address the following data and other policy-related requests:

1) ETHICS Request: In the methods section of your manuscript, please provide the identification number for the protocol approved by the ethics committee of the University of Lübeck. 

2) DATA Request: I noticed that in your Data availability statement you say "The complete dataset associated with this work will be publicly available under https://osf.io/28r57/". I was unable to find the data underlying this paper on osf, and so ask that you kindly direct me to it so that I can confirm that it meets our data sharing policy (apologies if I missed it). **IMPORTANT - please ensure that the data is accompanied by a README file that explains how the data were analyzed to generate the figures in the paper. Please also double check that the data meets our data sharing policy requirements: http://journals.plos.org/plosbiology/s/data-availability. For example, we will need you to provide the data underlying each figure in your study. As a final note, we request that each figure legend references the underlying data. For example, you might add the sentence "the data underlying this figure can be found at https://osf.io/28r57/" to each figure legend. 

3) TITLE: After discussion within the team, we think that the title might be edited slightly to make it more accessible to a broad audience. If you agree, we might suggest that you change the title to something like: “Dynamic large-scale connectivity between intrinsic cortical oscillations support adaptation during challenging listening conditions”. 

We expect to receive your revised manuscript within two weeks. 

*Published Peer Review History*

*Early Version*

Sincerely,

Lucas Smith, Ph.D.,

Associate Editor,

lsmith@plos.org,

PLOS Biology

Reviewer remarks:

Reviewer #1, Anne Keitel: The authors responded well to the comments and incorporated changes to make analyses and results even clearer. The answer to 'why these frequency bands', is not very specific, and does not explain why e.g. alpha was split into a higher and a lower band (although I think there is precedence and reason to do this, but this is not explained/justified in the manuscript). Also, there is one point that needs clarification, which is the reference to the Keitel & Gross 2016 study. This previous study found spectral modes between 1 and 120 Hz (most of which were in the alpha band), not "~7-30 Hz" as stated in the manuscript/reply. This should be amended.

I was also asked to assess the responses to concerns by Reviewer 2. Regarding their first point ("state-of-the-art", e.g. "network dynamics during crossmodal attentional shifts"), the authors now provide background on a few previous findings, although it is difficult to say whether these are what Reviewer 2 had in mind. Regarding the novelty aspect more broadly, the results probably do not seem surprising, because alpha and beta oscillations are commonly associated with attention and performance. However, the specific connectivity method (envelope correlations in EEG) in combination with the specific auditory behavioural paradigm yields a nice set of results.

Reviewer 2 also asked to explain the rationale for combining spatial and feature-based attention in the task. The authors elaborate on the use of spatial attention but do not explain why they combined this with feature-based attention.

The remaining points are answered satisfactorily, in my opinion.

Reviewer #3: The authors have made an excellent revision and addressed all my concerns. I have no further concerns left.

---

## [Editor Report · Decision Letter 3]

7 Sep 2021

Dear Dr Alavash,

On behalf of my colleagues and the Academic Editor, Jennifer Bizley, I am pleased to say that we can in principle offer to publish your Research Article "Dynamic large-scale connectivity of intrinsic cortical oscillations supports adaptive listening in challenging conditions" in PLOS Biology, provided you address any remaining formatting and reporting issues. These will be detailed in an email that will follow this letter and that you will usually receive within 2-3 business days, during which time no action is required from you. Please note that we will not be able to formally accept your manuscript and schedule it for publication until you have made the required changes.

**As you address these last formatting requests, we also ask that you address the following editorial requests: 

1) Thank you for providing the data underlying your supplemental figures. Will you please add a sentence to each relevant supplemental figure legend referencing this data? For example, you could add the sentence "The data underlying this figure can be found at https://osf.io/ge2cq/"

2) Thank you also for providing a description of the dichotic listening task as a supplemental file. We request that you move this into the main text as part of your materials and methods section. 

PRESS

Sincerely, 

Lucas Smith, Ph.D. 

Senior Editor 

PLOS Biology

lsmith@plos.org